# SPDER: Semiperiodic Damping-Enabled Object Representation

**Kathan Shah**
UC Berkeley
kathan@berkeley.edu

**Chawin Sitawarin**
UC Berkeley
chawins@berkeley.edu

## Abstract

We present a neural network architecture designed to overcome the spectral bias towards lower frequencies faced by conventional implicit neural representation networks. Our proposed architecture, SPDER, is a simple MLP that uses an activation function composed of a sinusoidal multiplied by a sublinear function, called the damping function. The sinusoidal enables the network to automatically learn the positional encoding of an input coordinate while the damping passes on the actual coordinate value by preventing it from being projected down to within a finite range of values. Our results indicate that SPDERs speed up training by $10\times$ and converge to losses $10{,}000-100{,}000\times$ lower than that of the state-of-the-art for image representation. SPDER is also state-of-the-art in audio representation. The superior representation capability allows SPDER to also excel on multiple downstream tasks such as image super-resolution and video frame interpolation. We provide intuition as to why SPDER significantly improves fitting compared to that of other INR methods while requiring no hyperparameter tuning or preprocessing. See code at https://github.com/katop1234/SPDER.

## 1 Introduction

Implicit neural representations (INRs) have demonstrated remarkable potential in applications such as novel view synthesis, image inpainting, super-resolution, and video frame interpolation (Sitzmann et al., 2020; Mildenhall et al., 2020; Chen et al., 2021; Simeonov et al., 2021; Park et al., 2019; Barron et al., 2021; Zhuang et al., 2022). Such representations, also known as coordinate-based multi-layer perceptrons (MLPs), can be modeled as

$$\phi_\theta(\mathbf{x}) \approx \psi(\mathbf{x}) \tag{1}$$

where $\phi_\theta$ is a neural network that learns a continuous mapping from a coordinate grid $\mathbf{x}$ to the ground truth signal $\psi$ of an object. Here, $x \in \mathbf{x}$ can represent a pixel coordinate in an image and $\psi(x)$ the pixel value at $x$.

These networks have attracted significant research interest for a number of reasons. Conventional methods for representing coordinate-based objects (e.g., images, videos, 3D objects, and audio) are restricted to a discrete input grid (Wallace, 1992; Painter and Spanias, 2000; Wiegand et al., 2003) while INRs can interpolate to continuous input values and thereby generate arbitrarily high resolution. They also encode an entire object solely within their weights, which is particularly useful due to their relative compactness (Strümpler et al., 2021; Yang et al., 2022; Ballé et al., 2016; Li et al., 2017).

However, coordinate-based MLPs, and neural networks collectively, have struggled with a spectral bias towards low-frequency modes and an underemphasis on center- and high-frequencies (Rahaman et al., 2018). They target lower frequencies, which tend to be more robust to perturbations, at the expense of higher ones encoding the finer structure of the signal. Consequently, they suffer from a greater loss than is ideal. Note that a bias towards lower frequency modes improves the generalization, whereas one towards higher frequencies improves task-specific performance (Cao et al., 2020; Xu et al., 2018).

Frequencies are fundamentally based on patterns in *values* over given *positions*. Consider an ideal INR: it should be capable of identifying that a pixel $P$ is located in the upper right corner of an image

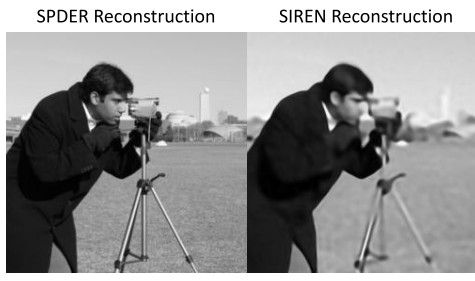
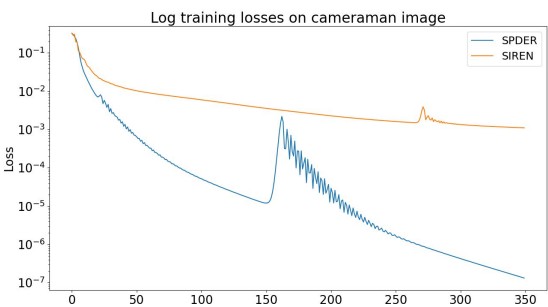

Figure 1: SPDERs learn neural representations of image magnitudes better than other MLP techniques. The 256x256 ground truth was trained on by SPDER (left) and SIREN (center), our baseline, for the same number of steps. By simply applying a damping factor to the sine nonlinearity, the loss goes down by nearly 12,000×, and training speeds up by 10×.

(position) and that $P$ represents part of the sky (value). Therefore, maintaining awareness of both positional and value information for inputs is crucial to overcoming spectral bias and also achieving good performance. This is important not only from a research standpoint, but also for applications such as reconstruction, compression, and edge detection. Our goal is to devise an INR architecture that maximizes

$$\rho\left(\mathcal{F}[\phi_\theta(\mathbf{x})], \mathcal{F}[\psi(\mathbf{x})]\right) \tag{2}$$

where $\rho$ is a function that returns a similarity score between two vectors and $\mathcal{F}$ is the Fourier transform (Morin, 2021). Recall that the Fourier transform provides an equivalent representation of an object over the frequency domain.

Conventional methods rely on a single MLP to represent an object. Here, each neuron technically has the receptive field of the entire object, which makes representing its complexity very challenging. Consequently, researchers map inputs through sinusoidals of certain frequencies during preprocessing to fit higher frequencies. However, this requires hardcoding inductive biases and does not necessarily capture the true spectrum (Ke et al., 2020; Yüce et al., 2021). Another technique, known as SIREN (Sitzmann et al., 2020), directly uses sine as the activation function of the MLP to help learn high frequencies. However, the output from a single neuron is "lossy" because of its many-to-one mapping. Hierarchical methods (Müller et al., 2022; Hao et al., 2022) break down an image into smaller chunks, for which less complex representations can be learned and then aggregated for the final output. Likewise, most neurons have much smaller receptive fields, which makes encoding complexity easier. These methods tend to have significant hardware requirements, which have prevented widespread adoption.

We propose SPDER to address all of the above. SPDER involves a single MLP with an activation function composed of a sinusoidal multiplied by a sublinear function, which we refer to as the *damping function*. We show that SPDER learns a near-perfect mapping from a coordinate grid to any high-fidelity signal including images, audio, and videos. This is because the sinusoidal enables learning the positional information of the input, while the damping helps pass on its value between layers. Note this allows the network to naturally learn a positional embedding.

We summarize the contributions of this paper below:

- We propose a simple and elegant architecture that captures the frequencies of different types of signals exceptionally well, achieving state-of-the-art in both image and audio representation. *Specifically, it outperforms state-of-the-art image INRs by several orders of magnitude. Compared to other non-hierarchical methods like itself, it speeds up training by 10× and converges to a 15,000× lower loss.*

- We demonstrate how our novel activation function helps a neural network overcome spectral bias and learn a positional embedding *without any hyperparameter tuning or hardcoded inductive biases* by having each neuron balance encoding both the position and value information of a signal.

- We highlight its generalizability through downstream applications such as image super-resolution, edge detection, video frame interpolation, and more.

## 2 RELATED WORK

Sitzmann et al. (2020) popularized general-use INRs by showing their ability to represent a myriad of objects such as images, audio, and 3D objects, as well as solve physics equations. SIREN, the architecture they proposed, is a simple MLP with sinusoidal activation functions and demonstrated a significant improvement over previous architectures. It is the main baseline we compare our architecture to.

Regarding balancing the position and value information of inputs, notably, Vaswani et al. (2017) found that hardcoding sinusoidals of logarithmically-scaled frequencies during preprocessing allowed a model to use the relative position of tokens in a sequence. This method is known as *positional encoding* (PE). Several INR architectures use a similar approach to represent higher frequencies (Mildenhall et al., 2020; Chen et al., 2021; Barron et al., 2021). However, these fail to successfully represent common media like images and audio without extra hyperparameter-tuning (Tancik et al., 2020; Ke et al., 2020; Luo et al., 2022).

The spectral bias of neural networks is also an interesting property we wish to highlight for this paper. Such a bias occurs because neural networks fit lower frequencies to generalize better and improve validation performance. However, they end up excluding the detailed, high-frequency structure of an input which leads to worse fitting (Xu et al., 2018; Rahaman et al., 2018). PE has been proposed to solve this, but interestingly, MLPs using PE have been shown to overfit the integer harmonics of the PE frequencies (Yüce et al., 2021). Moreover, they include hardcoding inductive biases.

Video-specific INR techniques were explored by (Chen et al., 2023; 2022; 2021). NeRF networks, used for novel view synthesis, typically use ReLU activations with sinusoidal positional encoding (Eq. (7)). Other methods, such as Fourier features, include appending the input with frequencies generated from a Gaussian distribution to overcome spectral bias (Tancik et al., 2020). These networks leverage neural tangent kernel theory (Jacot et al., 2020) and are widely used in computer vision. In general, sinusoidals have extensively been used in INR architectures because of their periodic nature and ability to map uniformly spaced inputs to a dense output signal.

Interestingly, Ramasinghe and Lucey (2021) examined several nonlinearities and concluded that a necessary condition for a neural implicit representation is having a nonlinearity with high Lipschitz smoothness. For example, Chng et al. (2022) demonstrated that Gaussian activation functions work under limited capacity.

Hierarchical methods have also shown great performance (Hao et al., 2022; Mildenhall et al., 2020). Notably, Instant-NGP (Müller et al., 2022) uses a hierarchy for an object over several resolutions, then queries a select few parameters to generate an output. It speeds up training for high-information media such as neural radiance fields, gigapixel images, and SDFs by several magnitudes. To the best of our understanding, Instant-NGP, in particular, learns a piecewise-linear approximation of an object and is only reasonable to use for extremely high-information objects like the ones listed prior. For example, in our experiments, we tried running Instant-NGP on several different machines, but all but a single high-end machine (costing over $20,000) failed due to lacking system requirements. Our method has no such restrictions and can even run on personal laptops.

## 3 APPROACH

### 3.1 MOTIVATION

Recall that for INRs, an input is a point from a coordinate grid, and the goal is to map it onto a signal (i.e. pixels of an image or samples of audio). An ideal network would correctly represent the frequencies between points in all directions of the grid while also keeping track of the actual coordinate value. This is because points along a direction may share similar frequencies/have dependencies, but have vastly different local features.

In SIREN, neurons massage points along the same frequencies onto the same outputs. Although they learn positional information automatically, they lose any sense of the original value of the coordinate point due to their mapping a wide domain onto a small, cyclical output range. For example, for a sinusoidal activation of period $0.25$, sample points $\{0, 0.25, 0.5\}$ with possibly vastly different local

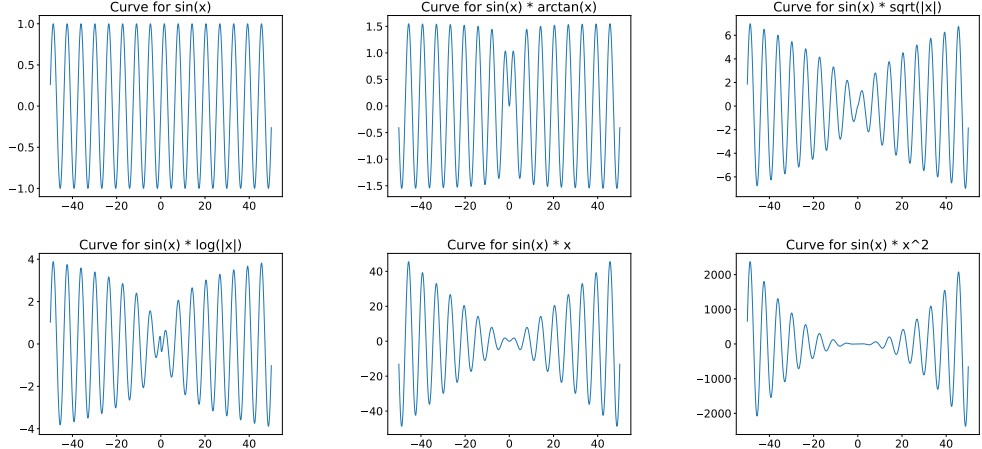

Figure 2: Six possible SPDER nonlinearities. Note for $\delta(x) = \arctan(x)$ (top center), the curve converges to $\pm\frac{\pi}{2}\sin(x)$ for $x$ sufficiently far from 0. For $\delta(x) = \sqrt{|x|}$ (top right), the magnitude of the curve grows relatively slowly, and it resembles two parabolas oriented sideways. For $\delta(x) = x$ (bottom center) and $\delta(x) = x^2$ (bottom right), we emphasize how the y-axis is significantly stretched out and the modulations drown out the periodicity. For these reasons, sublinear $\delta$ is desirable.

features would be mapped to the exact same output. To remember an input value, the network would thereby have to coordinate several neurons, reducing effectiveness.

## 3.2 DAMPING

SPDER is a neural network architecture that uses a *semiperiodic* activation function. Such functions are characterized as

$$\sin(x) \cdot \delta(x) \tag{3}$$

where $\delta$ is a sublinear function of $x$, also called the *damping function* (Fig. 2). We use the term *semiperiodic* to describe a function if it oscillates across the x-axis with a predictable pattern.

By modulating the magnitude of the sinusoidal through $\delta$, we embed knowledge of the raw value of input directly into the activations while still reaping the benefits of its positional information through the periodic sinusoidal. This acts as a substitute for artificial positional encoding but with no additional preprocessing.

For activation functions described by Eq. (3), the local Lipschitz constant is bounded by $\Theta(\delta(a) + \nabla\delta(a))$ for an input $a$ (proof in A.1). In plain English, this means the rate of change of the neuron's output with respect to small changes in its input is proportional to the damping function and its gradient evaluated at that input. Consequently, for linear (and superlinear) $\delta$, we empirically observed poor performance as the activation function exhibited unstable oscillatory behavior (Fig. 2). This meant the effective periods were far too irregular to encode meaningful frequencies. Restricting $\delta$ to be sublinear lowers the Lipschitz constant enough to ensure smoothness across all neurons and allows the network to balance both the input's positional and value information.

Unless stated otherwise, $\delta(x)$ will refer to the function $\sqrt{|x|}$, which is just the square root of $x$ extended to the negative numbers. Other promising damping functions that fared well on our tasks included $\log(|x|)$, $\arctan(x)$, and $\sqrt{\text{ReLU}(x)}$. We chose $\sqrt{|x|}$ because it performed exceptionally on our image dataset and has mostly well-defined gradients, but note the others may be valuable for different settings. For example, a SPDER with $\delta = \arctan$ represented audio best. We also note the case where $\delta$ is a constant is equivalent to SIREN which markedly underperforms against sublinear $\delta \in \Omega(1)$ like the ones listed above. We found SPDERs perform optimally in settings where they had to learn a mapping from a coordinate grid to a signal. For example, they are less successful at novel view synthesis, where the input direction is polar.

The jagged curve beyond step 150 in Fig. 1 indicates the complexity of the loss landscape that SPDER navigates. When optimizing past a certain loss threshold, SPDER must traverse a highly non-convex landscape, leading to fluctuations. Unlike other networks whose performance plateaus, SPDER's gradients can keep navigating the complex landscape towards the *global minimum*. In fact, SPDER's reconstruction often has 0 pixel deviations on an 8-bit scale. We note these loss spikes do not translate into visible defects in the resulting representations, and SPDER consistently outperforms all other architectures. It is certainly possible (but not necessary) to select the minimal loss representation up to a certain step or extend the training until the loss no longer spikes.

## 3.3 FREQUENCIES

In general, an implicit neural representation $\phi_\theta$ is fed in a *single* pair of input $\mathbf{x}$ and ground truth output $\psi(\mathbf{x})$ throughout training. The training loss is simply mean-squared error (MSE), i.e.,

$$\mathcal{L} = \|\phi_\theta(\mathbf{x}) - \psi(\mathbf{x})\|^2 \tag{4}$$

MSE, albeit a convenient objective for backpropagation, can be a misleading metric to humans for discerning the successful reconstruction of an object. In the case of images, for example, it is marginally affected by blurriness but is very sensitive to small shifts. Naturally, studying similarity in the frequency domain would mitigate these issues (Amir and Weiss, 2021; Zhang et al., 2018). Through Parseval's theorem, however, it can be shown that minimizing the MSE of the frequency domain of the represented image and ground truth is tantamount to minimizing the MSE between the pixels themselves (Wronski, 2021).

This necessitates finding another metric to validate the frequency structure of a representation. Referencing Eq. (2), we seek to maximize the cosine similarity between the amplitude spectrum of the neural representation and the ground truth. We formulate this as maximizing

$$\rho_{AG} = \frac{\sum_{n=0}^{N-1} A_n G_n}{\sqrt{\sum_{n=0}^{N-1} A_n^2} \sqrt{\sum_{n=0}^{N-1} G_n^2}} \tag{5}$$

where each $G_n$ is a zero-centered magnitude from the Fourier series of the ground truth signal we wish to represent, each $A_n$ is the corresponding zero-centered magnitude from the neurally represented signal, and $\rho_{AG}$ is the cosine similarity between $\mathbf{A} = (A_0, A_1, \ldots, A_{N-1})$ and $\mathbf{G} = (G_0, G_1, \ldots, G_{N-1})$. Note in Eq. (5), we omit any notion of phase shifts because the representation is learned from the same basis as that of the true signal, and hence, phase shifts should not impact the overall similarity. Therefore, for some vector $s$ encoding the signal of interest, the $n$-th term in $\mathbf{A}$ and $\mathbf{G}$ comes from the general formula for the Fourier amplitude spectrum of a discrete-time signal [1]:

$$\frac{2}{N} \left| \sum_{k=0}^{N-1} s_k e^{-i2\pi nk/N} \right| \tag{6}$$

We demonstrate how SPDER not only has incredibly low loss but also maximizes $\rho_{AG}$ and, therefore, captures the *exact* frequency structure of an image (Table 1). Moreover, it can achieve this in a compact and efficient architecture, with no hyperparameter- or fine-tuning. We believe this is because each neuron in a SPDER is designed to rapidly learn the frequency domain. Note we chose not to optimize for this metric during training as it's far more complicated to compute and had no practical advantage. We present it simply to lend insight into SPDER's overcoming of spectral bias.

We wish to highlight this as one of the most exciting features of our architecture, as significant research has been conducted on learning the frequency distribution of objects through positional encoding, hierarchies, increasing layer width, etc. These certainly help, but they introduce more inductive biases and most have yet to overcome the spectral bias known to pervade neural networks (Rahaman et al., 2018; Lieskovská et al., 2021; Bai et al., 2018). We find SPDER's performance to be extremely conceptually intriguing because it succeeds even when each neuron has the receptive field of the entire object. Therefore, we hope our findings can advance research done in fitting the frequency manifold of inputs.

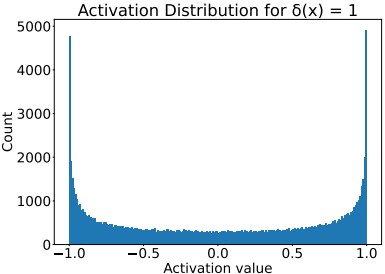 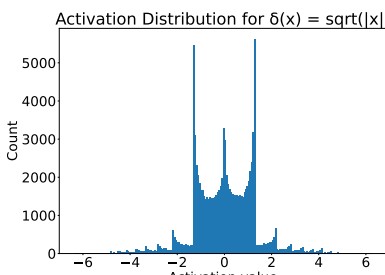

Figure 3: For $\delta(x) = 1$ (left), the activation values distinctly peak at local extrema of $\sin(x)$ at $\pm 1$. For $\delta(x) = \sqrt{|x|}$ (right), they peak at local extrema of $\sin(x) \cdot \sqrt{|x|}$ at $\pm 1.31, \pm 2.18, \pm 2.81, \pm 3.31$, etc. Using solely $\sin(x)$ as a nonlinearity discards information about the magnitude of the input, unlike in a conventional SPDER network.

### 3.4 ACTIVATIONS

Experimentally, we observed that the distribution of activation values across trained SPDERs peak at values of $x$ for which $\nabla(\sin(x) \cdot \delta(x)) = 0$ (Fig. 3, Fig. 8). Activated neurons apparently try to maximize the magnitude of their output by seeking out stationary points. This demonstrates feature selection as it means neurons conveying information have their signals amplified while others' outputs get minimized (similar to how a high-pass filter works).

Due to the damping, neurons with inputs of larger magnitude will, on average, be mapped to outputs of larger magnitude for $\delta \in \Omega(1)$. Moreover, given their semiperiodic nature, inputs spaced equally far apart will still be mapped to roughly equal outputs. Through maintaining this delicate balance, the network is able to relay value information from one layer to the next while automatically learning a positional embedding. Further examples in A.3.

## 4 RESULTS

### 4.1 BASELINES

**ReLU:** Standard activation for most MLPs. Neurons map an input $i$ to $\max(0, i)$.

**ReLU with positional encoding:** The input layer is appended with:

$$[\sin(\omega_0 i) \ \cos(\omega_0 i) \ \ldots \ \sin(\omega_{L-1} i) \ \cos(\omega_{L-1} i)] \tag{7}$$

where $\omega_k$ is the frequency of the $k$-th sinusoid, and $L$ is the number of frequency bands used for the encoding.

**ReLU with Fourier Feature Networks:** Similar to positional encoding, but frequencies are generated from a Gaussian distribution. We append 20 of such encodings to the input of a ReLU network from a standard Gaussian.

**Instant-NGP:** We show how although Instant-NGP converges very rapidly, it underperforms SPDER in the low-resolution regime. Further comments in A.8.

**SIREN:** Neurons map inputs to $\sin(\omega_0 i)$, where $\omega_0$ is a hyperparameter which is 30 unless stated otherwise.

**SPDER:** We demonstrate this to have a 10,000 to 100,000-fold improvement in image representation loss compared to SIREN. Neurons map inputs to $\sin(\omega_0 i) \cdot \delta(i)$ for sublinear $\delta$.

### 4.2 IMAGE REPRESENTATION

Results are shown in Table 1. SPDER consistently outperformed the other architectures across all metrics at all training steps. Specifically, we demonstrate how SPDERs excel at representing

---

[1]This can easily be evaluated through NumPy's `fft` module

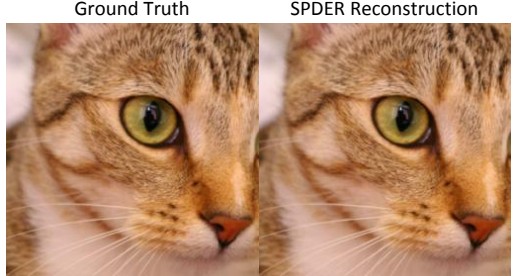 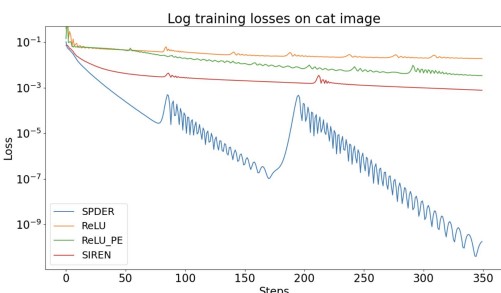

Figure 4: (Left) SPDER's reconstruction of the image from `skimage.data.cat()` after training for 500 steps (taking ∼55 seconds). Notice it has no aliases. (Right) The single-channel log-loss curves for various activation functions on the same image are shown. By the end, SPDER has more than 100,000× improvement compared to SIREN. Note that no configuration of positional encoding, network size, hierarchy, etc. from literature has been shown to match a simple SPDER on this task.

| Metric | Step | Architecture | | | | | |
|---|---|---|---|---|---|---|---|
| | | ReLU | ReLU w/ PE | ReLU w/ FFN | SIREN | Instant-NGP | SPDER |
| Loss (↓) | 25 | 0.1276±0.06 | 0.1204±0.05 | 0.2360±0.09 | 0.0334±0.02 | 0.0142±6e-3 | 0.0079±3e-3 |
| | 100 | 0.0850±0.05 | 0.0400±0.02 | 0.1325±0.06 | 0.0100±5e-3 | 0.0012±7e-4 | 7.0e-5±5e-5 |
| | 500 | 0.0498±0.03 | 0.0052±3e-3 | 0.1313±0.06 | 0.0010±5e-4 | 0.0001±9e-5 | **6.7e-8**±1e-7 |
| PSNR (↑) | 25 | 15.4±2.2 | 15.7±2.1 | 12.58±1.7 | 21.5±2.5 | 25.0±2.3 | 27.5±2.0 |
| | 100 | 17.4±2.5 | 20.7±2.6 | 15.29±2.2 | 26.9±2.9 | 36.2±2.9 | 48.6±3.1 |
| | 500 | 19.9±2.9 | 29.7±2.8 | 15.34±2.2 | 36.6±2.3 | 46.6±2.9 | **85.7**±10.7 |
| $\rho_{AG}$ (↑) | 25 | 0.7860±0.12 | 0.7823±0.12 | 0.5327±0.21 | 0.9522±0.03 | - | 0.9936±3e-3 |
| | 100 | 0.8543±0.10 | 0.9443±0.04 | 0.7208±0.14 | 0.9866±9e-3 | - | 0.99980±4e-4 |
| | 500 | 0.9185±0.06 | 0.9927±6e-3 | 0.7300±0.14 | 0.9988±9e-4 | - | **0.99995**±2e-4 |

Table 1: **DIV2K 256×256 image representation loss, peak signal-to-noise ratio (PSNR), and frequency similarity ($\rho_{AG}$) with standard deviations over various training steps.** Sample sizes: ReLU (N=801), ReLU w/ PE (N=800), ReLU w/ FFN (N=800), SIREN (N=801), Instant-NGP (N=800), and SPDER (N=702). Note how by 500 steps in, SPDER's representation is virtually lossless (A.13). The PSNRs are also significantly above that of SIREN's and Instant-NGP's, which both have decent quality (A.14). We emphasize how within merely 25 steps, SPDER's loss is low enough such that the representation is visually pleasing to any viewer. SPDER has near-perfect cosine similarity with the amplitude spectrum of the images, which means it faces no difficulty overcoming the spectral bias known to pervade conventional neural networks.

images, with a 10× improvement in training steps to achieve a similar loss to our baselines and 10,000-100,000× reduction in loss compared to the others. The amplitude correlation of SPDER's representation with the ground truth is greater than 0.9999, indicating an almost perfect representation of the frequencies. A PSNR of 45 is generally considered excellent quality, and SPDER achieves significantly beyond that. We observed that in most cases, SPDER's representation exactly matched the ground truth after 500 steps (A.13). It is the *only* known method in literature to achieve lossless neural image representation (which we claim to be more proof it overcomes spectral bias for images).

We also wish to reinforce the sheer simplicity of the model. Being solely a 5-layer MLP, one could initialize it in PyTorch in less than 20 lines of code and achieve these results.

### 4.2.1 IMAGE GRADIENT REPRESENTATION

We observed that SPDER is able to learn a near-perfect representation of the gradients of an image $I$ (Fig. 5). SPDER models are supervised solely on the pixel values of an image, and the gradients are determined from their reconstruction of the image. Consequently, the model could only succeed in this task if its representation accurately included the structure of the edges.

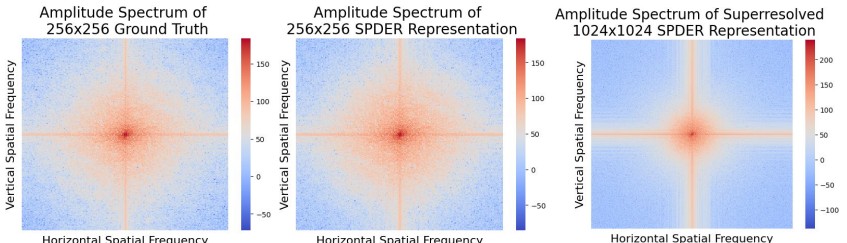

Figure 5: 5-layer networks were trained on the image from `skimage.data.text()` (1st from left). The gradients of their reconstructions are shown. Note how SPDER represents the edges well within merely 10 training steps (2nd), especially w.r.t. SIREN and ReLU with positional encoding (4th, 6th). After 25 steps, SPDER has captured the exact edges of the image (3rd).

Figure 6: 2D amplitude spectrums of a single channel of the $256\times256$ ground truth image (left) from `skimage.data.astronaut()` and of SPDER's representation of it (center) are shown. While interpolating to resolution 1024x1024 (right), SPDER is able to preserve the structure of the original signal, hence the defined horizontal and vertical bands near the center, which represent lower frequencies. Note the solid, pale blue regions around the corners of the 1024x1024 resolution spectrum, which indicate SPDER introduces minimal high-frequency content/noise into the super-resolved image.

Properties of the Fourier transform dictate that determining $\mathcal{F}(I)$ and the frequency coordinate along one axis uniquely determines $\mathcal{F}$ of the gradient of $I$ along that axis. This is formulated as:

$$\mathcal{F}(\nabla_y I) = i \cdot \omega_y \cdot \mathcal{F}(I) \tag{8}$$

where $\omega_y$ is the frequency in the y-direction and $i$ is the imaginary unit (proof in A.2). Therefore, since SPDER learns the correct pixel frequency structure, it is also able to recognize the edges properly. Further examples in A.4.

### 4.3 SUPER-RESOLUTION

While super-resolving images, SPDER preserves the frequency structure of the training image while introducing only the minimal necessary higher frequencies required for smooth interpolation (Fig. 6). One might believe that a chaotic function like $\sin(x) \cdot \sqrt{|x|}$ or $\sin(x) \cdot \log(|x|)$ oscillates far too irregularly to interpolate meaningfully between points it has been trained on; however, we've empirically determined this is not the case (Fig. 7). Further details and examples in A.5.

### 4.4 AUDIO REPRESENTATION AND INTERPOLATION

To the best of our knowledge, SPDER's performance is state-of-the-art for neural audio representation. Interestingly, for our experiments on audio, we found the optimal $\delta$ to be $\arctan$. Our findings are in Table 2. We also test SPDER's performance in audio interpolation. Further details in A.6.

### 4.5 VIDEO REPRESENTATION AND FRAME INTERPOLATION

We show how SPDER can represent video as well, achieving PSNRs of $\sim 30$, which indicate good quality reconstructions. We then asked our overfit SPDER network to interpolate to frames it hadn't directly been trained on, and it could do so with plausible results. This is exciting because it indicates that SPDER is learning the true frequency structure of the video. For example, if it were biased towards lower frequencies, the interpolations would be blurry or blocky. If the bias was towards higher frequencies, we would see grainy frames. Further details and examples in A.7.

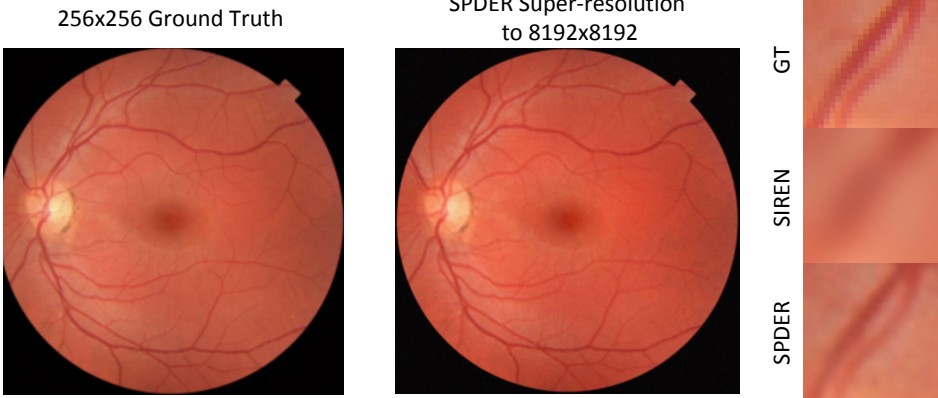

Figure 7: `skimage.data.retina()` at 256x256 ground truth (left) and SPDER super-resolution of it to 8192x8192 (center). Note this has $1024\times$ pixels, and yet SPDER does not introduce any additional noise. Moreover, the output image has $\sim$201 million parameters, while the SPDER has less than 330,000 parameters (a 99.84% reduction). The detail around a blood vessel is shown for the ground truth (right above). A SIREN and SPDER are trained for the same number of steps on the ground truth and then asked to super-resolve to 8192x8192 (right center and below). Without any priors, SPDER is still visibly clearer and creates a plausible reconstruction.

| Metric | Step | Architecture | | | | |
| | | ReLU | ReLU w/ FFN | ReLU w/ PE | SIREN | SPDER |
|---|---|---|---|---|---|---|
| Loss ($\downarrow$) | 25 | 0.3718±0.56 | 0.3694±0.56 | 0.0298±0.04 | 0.0172±0.03 | 0.0135±0.02 |
| | 100 | 0.0294±0.04 | 0.0294±0.04 | 0.0291±0.04 | 0.0084±0.02 | 0.0052±0.01 |
| | 500 | 0.0292±0.04 | 0.0292±0.04 | 0.0284±0.04 | 0.0021±9e-3 | 0.0009±3e-3 |
| | 1000 | 0.0291±0.04 | 0.0291±0.04 | 0.0279±0.04 | 0.0007±4e-3 | **0.0003**±2e-3 |
| $\rho_{AG}$ ($\uparrow$) | 25 | 0.0378±0.09 | 0.0388±0.10 | 0.0947±0.10 | 0.5740±0.30 | 0.7276±0.25 |
| | 100 | 0.0928±0.11 | 0.0934±0.12 | 0.1144±0.11 | 0.7744±0.27 | 0.8843±0.19 |
| | 500 | 0.1026±0.12 | 0.1032±0.12 | 0.1944±0.12 | 0.9288±0.17 | 0.9699±0.09 |
| | 1000 | 0.1031±0.12 | 0.1037±0.12 | 0.2676±0.13 | 0.9647±0.12 | **0.9863**±0.06 |

Table 2: **ESC-50 audio average representation loss and frequency similarity ($\rho_{AG}$) with standard deviations over various training steps.** Sample sizes: ReLU (N=779), ReLU w/ FFN (N=787), ReLU w/ PE (N=747), SIREN (N=795), and SPDER (N=637). We note that the majority of correlations we recorded for SPDER by step 1000 were above 0.99, indicating near-perfect reconstruction of the audio (see A.6.2).

## 5    CONCLUSION AND FUTURE WORK

In this work, we presented a simple and intuitive MLP architecture that can be used to represent images with 10,000-100,000$\times$ lower loss than current state-of-the-art methods without the need for priors or augmentations. We show its generalizability in representing audio, video, gradients of images, etc. We also believe our architecture's automatic balancing of the positional and value information of inputs can offer new insights into learned positional embeddings. Likewise, studying how SPDER overcomes spectral bias may help us learn how to improve neural network performance generally. Through our interpolation experiments, we also show that SPDER preserves the frequency spectrum for points it hasn't seen and, in a sense, has low spectral "variance" as well.

The most reasonable next steps include combining hierarchical methods with SPDER, where the network learns each small block extremely accurately, then aggregates them for a near-perfect representation efficiently. Eventually, as neural hardware becomes more commonplace and advanced, the use of neural networks to compress and represent media will become increasingly attractive. The ability for INRs to capture frequency structure efficiently will lead to a whole host of applications where information is stored in the weights of a network rather than in bytecode.

ACKNOWLEDGEMENTS

The second author of this paper was supported in part by funds provided by the National Science Foundation (under grant 2229876), the KACST-UCB Center for Secure Computing, the Department of Homeland Security, IBM, the Noyce Foundation, Google, and Open Philanthropy. Any opinions, findings, and conclusions or recommendations expressed in this material are those of the author(s) and do not necessarily reflect the views of the sponsors.

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

# A  Supplementary Material

## A.1  Proof of Lipschitz bound

As stated in the main paper, for activation functions described by the expression in Eq. (3), the local Lipschitz constant is bounded by $\Theta(\delta(a) + \nabla\delta(a))$ in the neighborhood around an input $a \in \mathbb{R}$, when $\delta$ is differentiable. We demonstrate this by deriving the lower and upper bounds of the derivative of $\sin(x) \cdot \delta(x)$ and showing equality between the two. For the proof, we will assume that $\delta$ is differentiable across all points barring a measure zero set, which means in practice, we are not concerned with singularities in $\nabla\delta$ since they are highly unlikely to occur.

*Proof.* Let $L(a)$ be the local Lipschitz constant of a nonlinearity around an input point $a$. We are interested in $L$ evaluated within a neighborhood $N$ of $a$ as it is a good indicator of if the function oscillates too much for any given input. Ideally, $L$ would grow slowly. Note that bounding $L$ is tantamount to determining the magnitude of the derivative of the nonlinearity.

$$L(a) = \max_{x \in N(a)} |\nabla(\sin(x) \cdot \delta(x))| \tag{9}$$

$$= \max_{x \in N(a)} |\cos(x) \cdot \delta(x) + \sin(x) \cdot \nabla\delta(x)| \tag{10}$$

$$\leq \max_{x \in N(a)} |\cos(x) \cdot \delta(x)| + |\sin(x) \cdot \nabla\delta(x)| \tag{11}$$

Since both $\cos(x)$ and $\sin(x)$ are bounded by $[-1, 1]$, we can write

$$L(a) \leq \max_{x \in N(a)} |\delta(x)| + |\nabla\delta(x)| \tag{12}$$

Note that the absolute value operator simply applies a constant factor to an input and that the derivative of a differentiable function is bounded by a constant factor over a sufficiently small $N(a)$. Therefore, we conclude that $L(a) \in \mathcal{O}(\delta(a) + \nabla\delta(a))$.

Likewise, we wish to determine the lower bound. Since both $\cos(x)$ and $\sin(x)$ are bounded by $[-1, 1]$, we can assign them to the constant values $c_1$ and $c_2$, respectively.

$$L(a) = \max_{x \in N(a)} |\cos(x) \cdot \delta(x) + \sin(x) \cdot \nabla\delta(x)| \tag{13}$$

$$= \max_{x \in N(a)} |c_1 \cdot \delta(x) + c_2 \cdot \nabla\delta(x)| \tag{14}$$

In the worst-case scenario where either $c_1$ or $c_2$ is 0, we have that

$$L(a) \geq \max\left(|c_1 \cdot \delta(a)|, |c_2 \cdot \nabla\delta(a)|\right) \tag{15}$$

We note that event that this occurs with a probability measure of 0, but lower bounds all other scenarios. Note that $\max(a, b) \in \mathcal{O}(a + b)$. Therefore, it implies $L(a) \in \Omega(\delta(a) + \nabla\delta(a))$. Because both the lower and upper bounds are equal, the Lipschitz constant $L$ evaluated near a point $a$ is in $\Theta(\delta(a) + \nabla\delta(a))$.

$\square$

## A.2  Proof of Relation between Fourier Transform of an Image and its Gradient

The proof is adapted from Gonzalez (2013).

*Proof.* Let $f(t)$ be a signal with Fourier transform $F(\omega)$. In the frequency domain, the signal $f(t)$ can be represented as follows:

$$f(t) = \frac{1}{2\pi} \int_{-\infty}^{\infty} F(\omega)e^{i\omega t}d\omega \tag{16}$$

Applying the gradient operator to $f(t)$, we obtain:

$$\nabla_t f(t) = \nabla_t \left( \frac{1}{2\pi} \int_{-\infty}^{\infty} F(\omega)e^{i\omega t}d\omega \right) = \frac{1}{2\pi} \int_{-\infty}^{\infty} i\omega F(\omega)e^{i\omega t}d\omega \qquad (17)$$

Therefore, the Fourier transform of $\nabla f(t)$ is $i\omega F(\omega)$, establishing the desired relationship between the Fourier transform of a signal and its gradient. This implies that knowing the frequency representation of a signal is enough to deduce the frequency representation of the gradient of that signal. $\qquad \square$

## A.3 ACTIVATIONS

As can be seen in Fig. 8, the outputs of the linear layers are normally distributed. This phenonemon was originally observed to occur for SIREN, where Sitzmann et al. (2020) claim that this proves how the architecture does not suffer from vanishing or exploding gradients.

The outputs of the activations in each layer cluster around the stationary points of $\sin(x) \cdot \sqrt{|x|}$. Note how in the earlier layers of the network, the activations are spread over a wide domain, but by the later layers, they are concentrated in a few stationary points near $x = 0$. We speculate that this demonstrates "learning"/feature selection, where the network first takes in a wide range of frequencies and then aggressively distills it into the most important components.

### A.3.1 ACTIVATION DISTRIBUTIONS FOR OTHER $\delta$'S

We include the distribution of activations for $\delta(x) = \log(|x|)$ (Fig. 9) and $\delta(x) = \arctan(x)$ (Fig. 10). When the $\delta$ has a large span, these activations can peak at many values, and therefore be somewhat representative of the magnitude of the input that went into it. For example, in Fig. 9, an input into a neuron has many possible "peaks" to choose from, so if it has a higher input magnitude, it will likely be mapped to a higher output magnitude. In Fig. 10, it's mostly going to be either 0 or 1.04, and the exact input that went into it would be very hard to recover. This is because $\sin(x) \cdot \arctan(x)$ has distinct stationary points near the origin, which then all converge to constant values sufficiently far from 0. However, the stationary points for $\sin(x) \cdot \log(|x|)$ keep growing without bound.

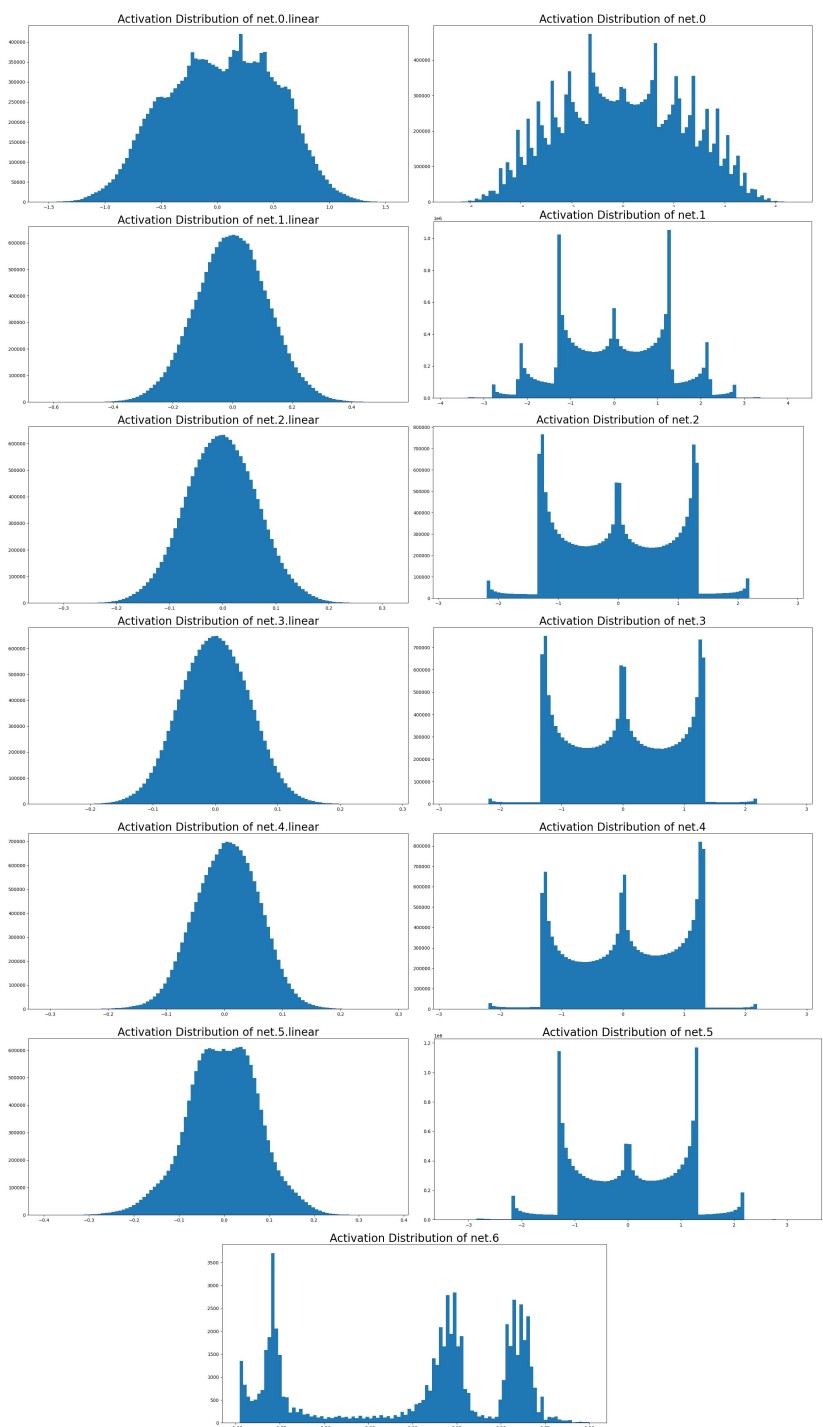

Figure 8: Distribution of activations of each layer of a 5-layer SPDER with $\delta(x) = \sqrt{|x|}$ trained on `skimage.data.camera()`. The output of the linear layers are on the left, and the outputs from the neurons in each layer are on the right, which peak at the stationary points of $\sin(x) \cdot \sqrt{|x|}$ (notice their resemblance to a spider, the architecture's namesake). The activations of the final output are shown at the bottom.

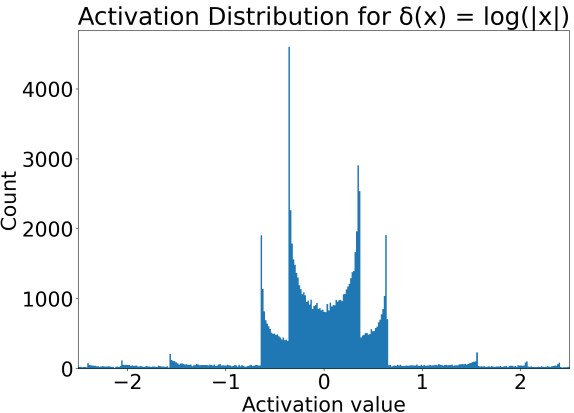

Figure 9: The activation values distinctly peak at local extrema of $\sin(x) \cdot \log(|x|)$ at $\pm 0.36$, $\pm 0.64$, $\pm 1.56$, $\pm 2.06$, etc.

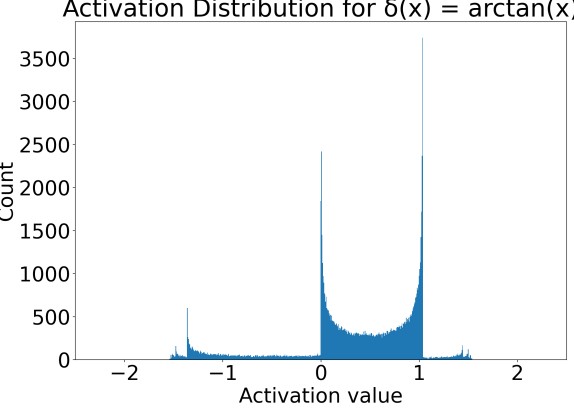

Figure 10: The activation visibly peaks at 0 and 1.04 here, although the stationary points of $\arctan(x) \cdot \sin(x)$ technically approach $-\frac{\pi}{2}$ and $\frac{\pi}{2}$ in either limit. This is because the stationary points nearest $x = 0$, where most of the inputs to neurons are, occur at 0 and 1.04. Beyond that, they're distributed extremely close to $-\frac{\pi}{2}$ and $\frac{\pi}{2}$. See Fig. 2 for a visual of the activation.

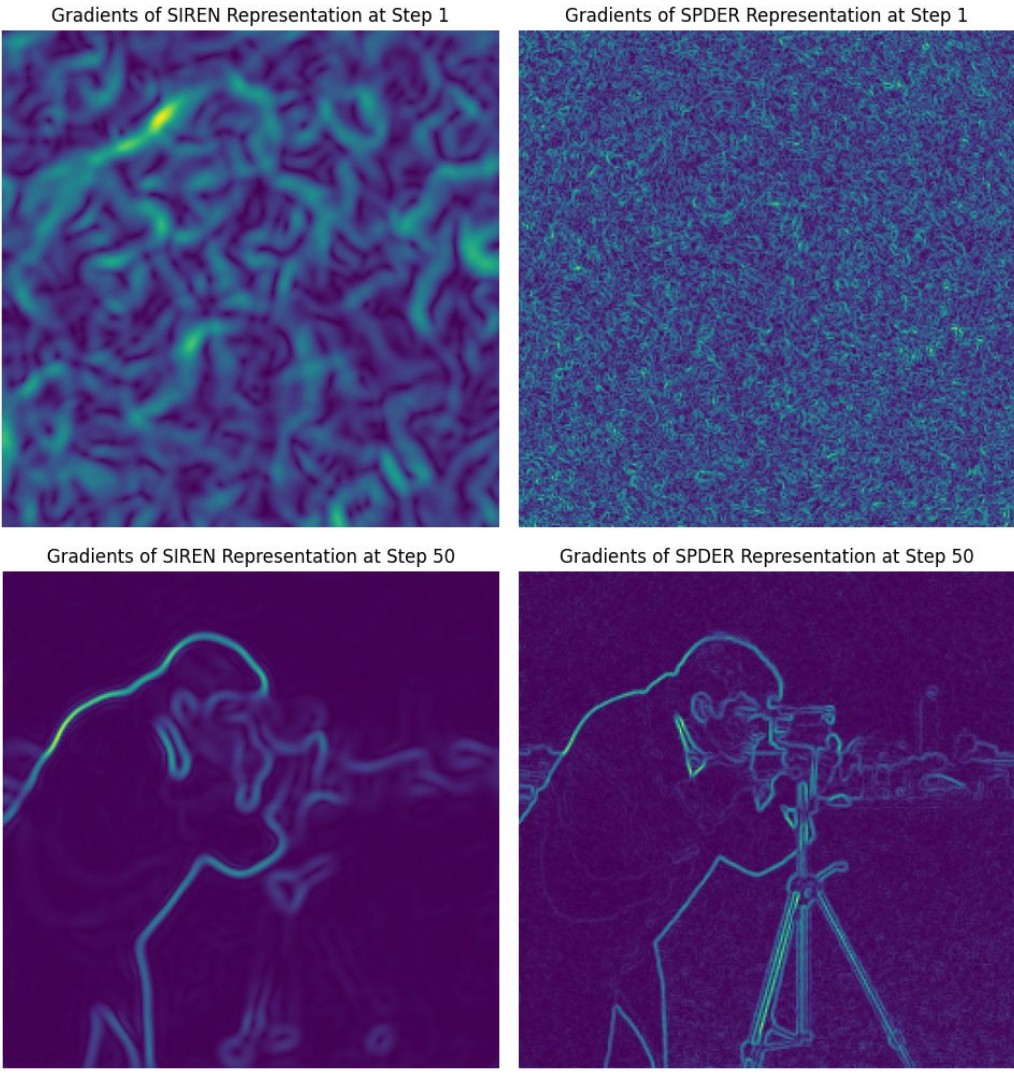

Figure 11: The gradients of the representations after training on the image from `skimage.data.camera()` for a *single* step are shown (top). Interestingly, using only a sine activation (top left) yields a few, thick gradients in the beginning. Adding a damping factor (top right) enables the representation to start off with hundreds of small strands which can presumably align around detail at any frequency much more accurately. We believe this is because of the semiperiodic nature of the activation function, where a *continuum* of frequencies can be encoded by a single neuron, whereas sinusoidals encode some number of *discrete* frequencies.

## A.4 IMAGE GRADIENT REPRESENTATION

Visuals are in Fig. 11 and Fig. 12. The intensity of each pixel in the resulting gradient image corresponds to how much a small change in each input coordinate value would affect the corresponding pixel prediction. Areas such as edges and other high-frequency components exhibit high gradient values.

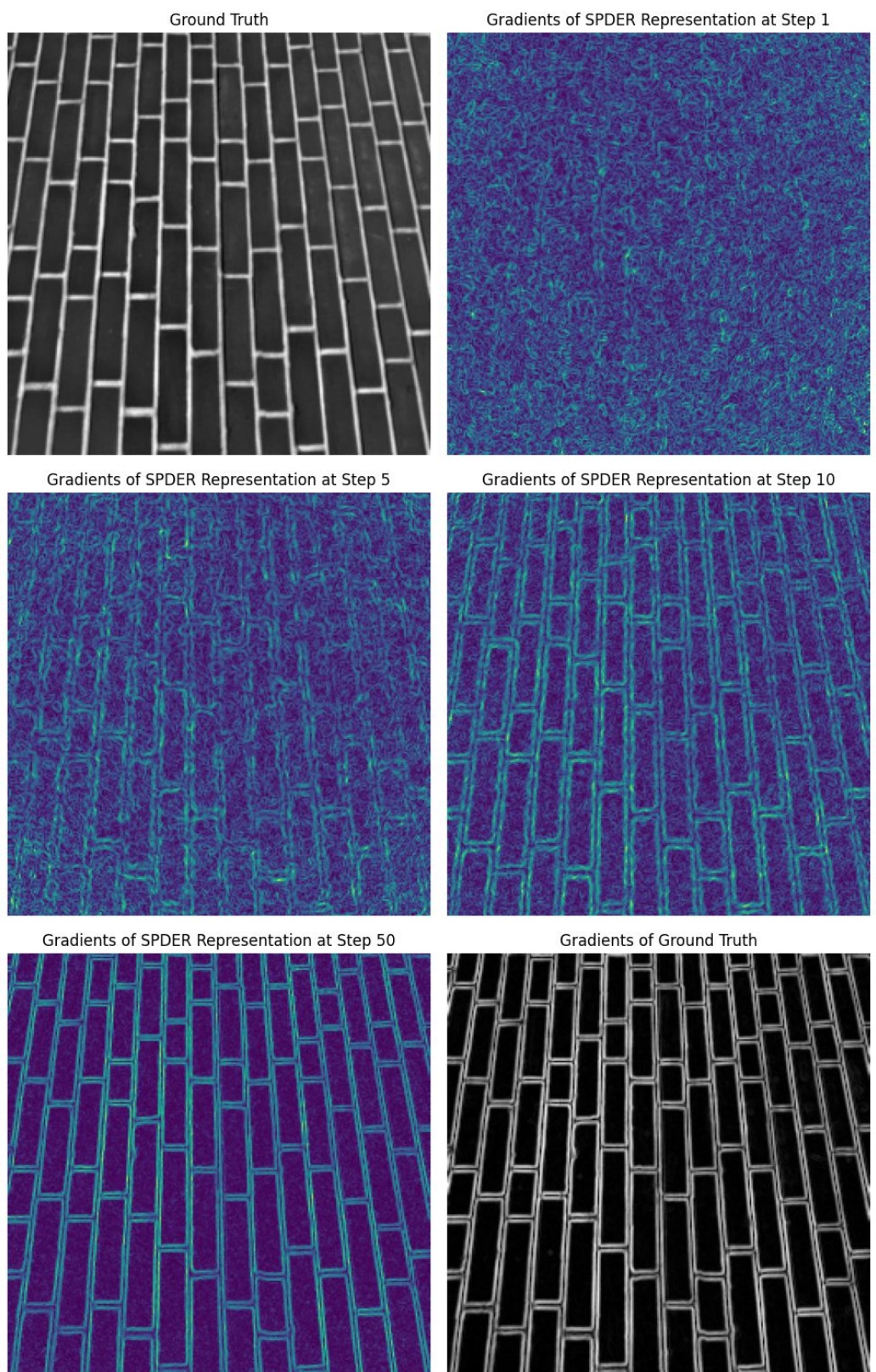

Figure 12: Gradients of SPDER representation of the image from `skimage.data.brick()` over various training steps. Note how SPDER can encode straight edges (low frequencies) incredibly well.

| Metric | Dataset | Super-resolution Factor (SRF) | | | | | |
| | | 2x | | 4x | | 8x | |
| | | SIREN | SPDER | SIREN | SPDER | SIREN | SPDER |
|---|---|---|---|---|---|---|---|
| MSE ($\downarrow$) | DIV2K | 2e-2±8e-3 | 6e-3±3e-3 | 2e-2±9e-3 | 7e-3±3e-3 | 2e-2±9e-3 | 8e-3±3e-3 |
| | ffhq0 | 9e-3±4e-3 | 3e-3±1e-3 | 8e-3±3e-3 | 3e-3±1e-3 | 9e-3±3e-3 | 3e-3±1e-3 |
| | ffhq1 | 9e-3±5e-3 | 4e-3±3e-3 | 8e-3±5e-3 | 3e-3±3e-3 | 8e-3±5e-3 | 4e-3±3e-3 |
| | ffhq2 | 1e-2±8e-3 | 4e-3±2e-3 | 1e-2±8e-3 | 3e-3±1e-3 | 1e-2±8e-3 | 3e-3±2e-3 |
| | Flickr2K | 4e-2±3e-3 | 1e-2±9e-3 | 4e-2±3e-2 | 2e-2±1e-2 | 4e-2±3e-3 | 2e-2±1e-2 |
| PSNR ($\uparrow$) | DIV2K | 23.2±1.9 | 29.4±2.6 | 22.8±2.0 | 28.0±2.4 | 22.8±2.0 | 27.7±2.3 |
| | ffhq0 | 26.7±1.6 | 31.6±2.0 | 27.0±1.6 | 32.2±2.0 | 27.0±1.6 | 31.8±1.9 |
| | ffhq1 | 27.3±2.3 | 31.2±2.5 | 27.8±2.4 | 31.9±2.5 | 27.7±2.4 | 31.4±2.6 |
| | ffhq2 | 25.3±2.7 | 30.5±2.1 | 25.8±2.7 | 31.5±1.9 | 25.7±2.7 | 31.0±2.0 |
| | Flickr2K | 23.2±2.0 | 25.6±3.0 | 20.8±2.7 | 25.1±2.8 | 20.8±2.7 | 24.9±2.8 |
| $\rho_{AG}$ ($\uparrow$) | DIV2K | 0.99165±6e-3 | 0.99751±3e-3 | 0.99110±6e-3 | 0.99712±3e-3 | 0.99108±6e-3 | 0.99702±3e-3 |
| | ffhq0 | 0.99799±1e-3 | 0.99942±4e-4 | 0.99823±1e-5 | 0.99956±3e-4 | 0.99819±1e-3 | 0.99953±3e-4 |
| | ffhq1 | 0.99835±8e-4 | 0.99949±2e-4 | 0.99856±7e-4 | 0.99962±2e-4 | 0.99852±7e-4 | 0.99958±2e-4 |
| | ffhq2 | 0.99775±1e-3 | 0.99920±5e-4 | 0.99812±1e-3 | 0.99945±3e-4 | 0.99807±1e-3 | 0.99941±3e-4 |
| | Flickr2K | 0.96332±3e-2 | 0.98891±1e-2 | 0.96324±4e-2 | 0.98825±1e-2 | 0.96320±4e-2 | 0.98797±1e-2 |

Table 3: **Super-resolution metrics with standard deviations over SRFs of $2\times$, $4\times$, and $8\times$.** The base resolution was 512x512, and the super-resolutions were at 1024x1024, 2048x2048, and 4196x4196, respectively. We chose three different subsets of the Flickr-Faces-HQ dataset (each corresponding to a different category and, therefore, distribution) denoted as ffhq0, ffhq1, and ffhq2. As stated before, by simply applying a damping factor, the super-resolution quality goes up significantly. Note how the PSNRs are ∼5 points higher compared to SIREN in each subcategory. For implementation details, see A.9.4.

## A.5  SUPER-RESOLUTION

In this section, we wish to highlight one application of SPDER's success in capturing the frequency structure: generating higher resolution detail **without the need for priors or meta-learning**.

We noticed that when interpolating from a base resolution of 512x512 or higher, no noise visible to the naked eye is introduced into any segment of the super-resolved reconstruction at *any* super-resolution factor. At base resolutions below that, some noise is visible when interpolating by $4\times$ or more, presumably because the SPDER has too few points to meaningfully learn the frequency distribution. We note this is still impressive, given that SPDER networks have no outside prior on pixel distributions.

### A.5.1  RESULTS

We include quantitative results of the comparison between SPDER and SIREN in Table 3. Interestingly, SPDER can superresolve to a factor of $8\times$ ($64\times$ pixels) and still have high reconstruction quality consistently. Most of the PSNRs of SPDER's super-resolutions are above 30, indicating good quality. We once again highlight how simply applying a damping factor to a nonlinearity boosts performance noticeably.

### A.5.2  EXAMPLES

Visuals are in Fig. 13 and Fig. 14.

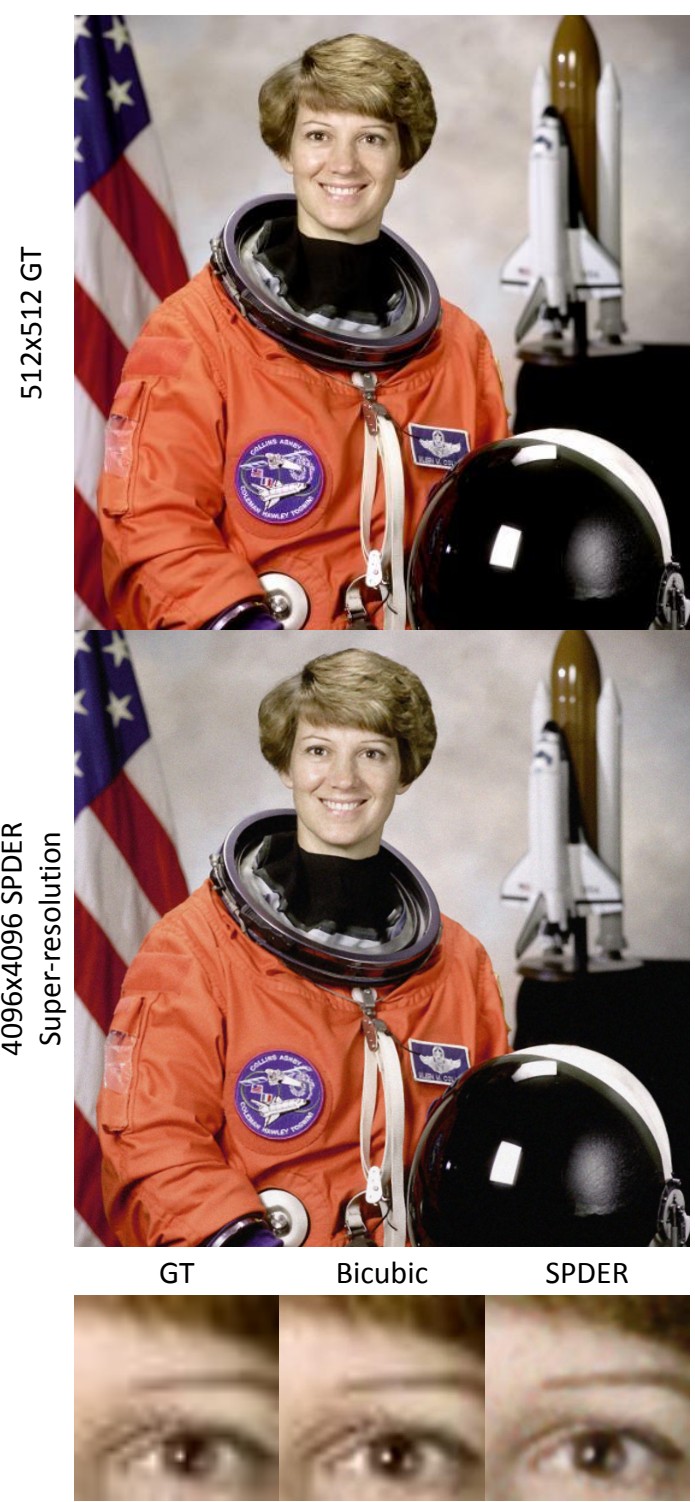

Figure 13: Image from `skimage.data.astronaut()` at 512x512 ground truth (top) and SPDER super-resolution of it to 4096x4096 (center). (Bottom, left to right) The detail around the eye is shown at the 512x512 ground truth, bicubic interpolation to 4096x4096, and SPDER's super-resolution to 4096x4096, respectively. Note how compared to bicubic, SPDER is visually sharper and traces out the eyelids better.

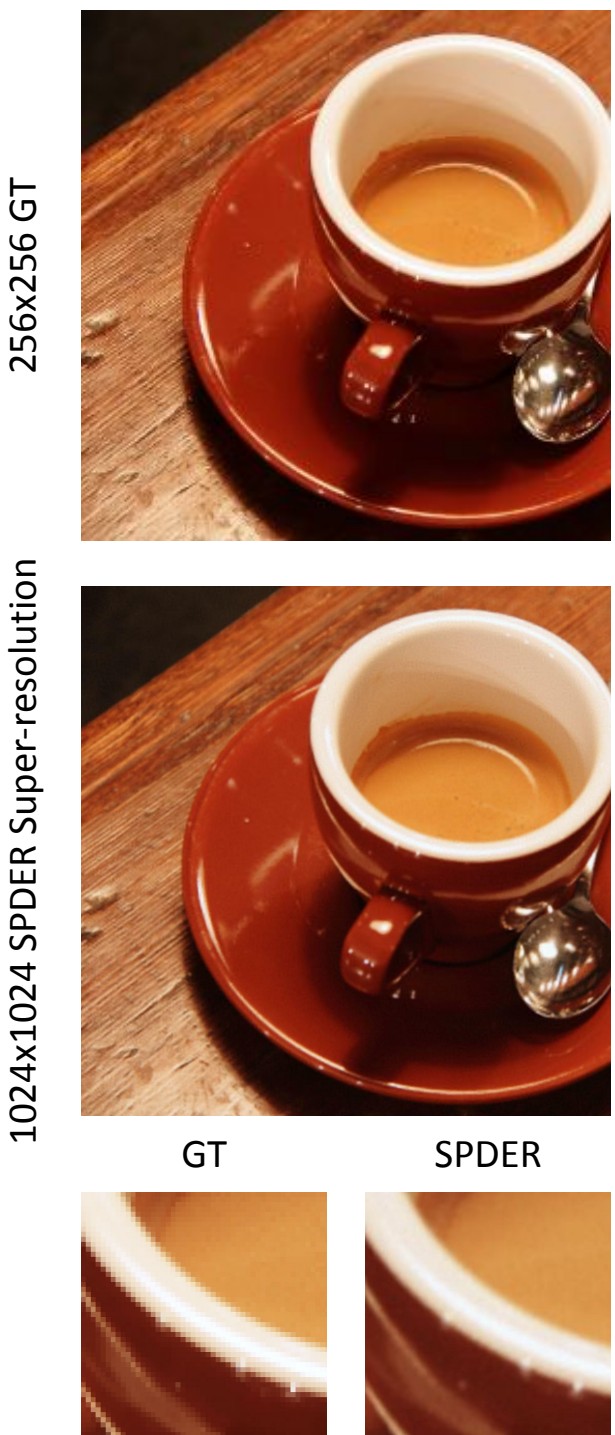

Figure 14: Image from `skimage.data.coffee()` at ground truth resolution 256x256 (top) and SPDER super-resolution to 1024x1024 (center). Edge of cup for ground truth (bottom left) and SPDER super-resolution (bottom right) are also shown.

| Metric | UF | Architecture | |
| --- | --- | --- | --- |
| | | SIREN | SPDER |
| | 2x | 0.00719±0.014 | **0.00522**±0.012 |
| MSE (↓) | 4x | 0.00959±0.017 | **0.00755**±0.015 |
| | 8x | 0.00959±0.017 | **0.00874**±0.017 |

Table 4: **Audio interpolation results: mean-squared error over various audio upsampling factors (UFs) with standard deviations over UFs of 2×, 4×, and 8×.** Across all upsampling factors, SPDER's mean reconstruction error is lower than that of SIREN's.

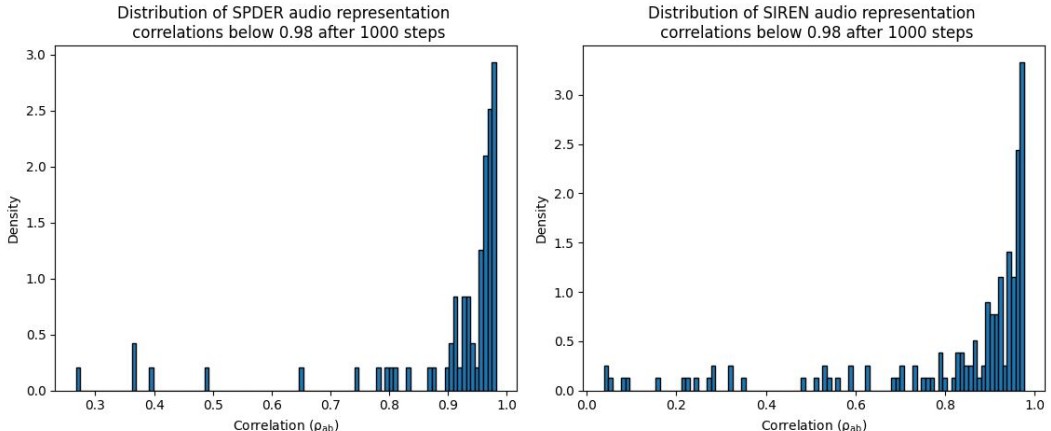

Figure 15: **Distribution of $\rho_{ab}$'s below 0.98 for SPDER and SIREN audio representations.** We drop the top 2 out of 100 bins for both SPDER and SIREN (dropping 89.4% and 81.4% of the total count, respectively) to make these bars visible. Note how SPDER has a noticeably sparser left tail, meaning it can represent many challenging examples SIREN cannot.

## A.6 AUDIO REPRESENTATION AND INTERPOLATION

### A.6.1 INTERPOLATION

Here, we show SPDER's metrics on reconstructing audio clips even after being trained on a small fraction of the entire clip. This is promising because it demonstrates that applying damping indeed preserves the frequency structure not just for 2D images but 1D audio better than without having it.

Quantitative results are in Table 4. We train an INR on an 8× downsampled version of an entire ground truth audio clip. The model then has to predict the values at the upsampling factors (UFs) of 2×, 4×, and 8×, where the UF is how many times larger the inference sample is than the training one. For example, the 8× upsampled version is simply the ground truth (GT), 4× UF corresponds to GT downsampled by 2×, and 2× UF is GT downsampled by 4×. See further implementation details in A.9.6.

### A.6.2 AUDIO DISCUSSION

We observed that in both SIREN and SPDER, the vast majority of reconstructions had a high correlation with the ground truth, with the peak in the 0.99-1.00 bin. Both architectures also had a handful of pathological samples they were not able to reconstruct accurately. However, the distribution of $\rho_{AG}$ for SIREN has a noticeably fatter left tail, meaning it has much more difficulty representing some samples that SPDER has no problem with (Fig. 15).

In general, the performance on audio was strongly impacted by a small percentage of pathological (highly noisy) outliers in our dataset. We chose *not* to exclude these from our results to keep the data consistent. However, in the real world, assuming the audio clips are intelligible and not full of noise,

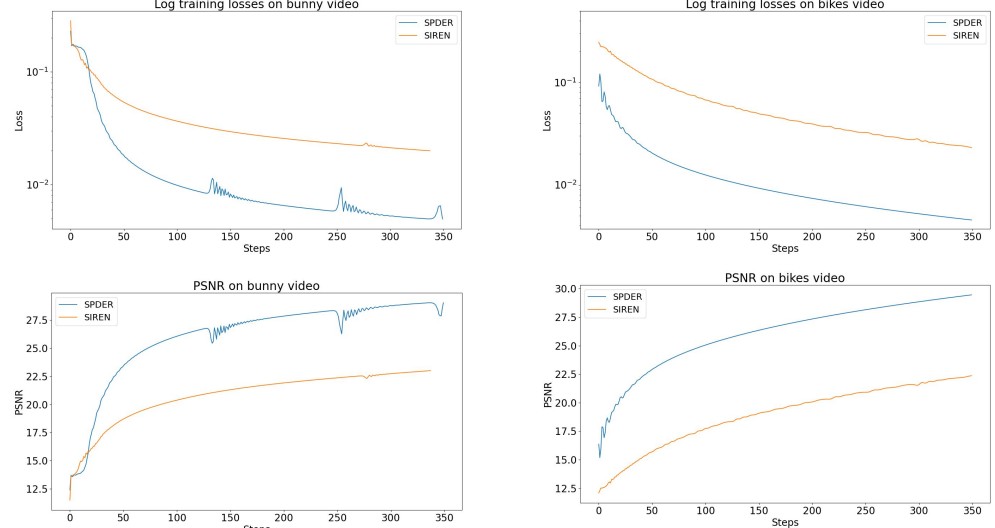

Figure 16: **Log training loss and PSNR curves for two videos for SPDER and SIREN.** Note how SPDER's PSNRs are consistently ∼9 points above that of SIREN's (bottom left and right). This illustrates how the simple idea of damping can succeed on non-spatial features (i.e. video frames). There is a performance difference between the two examples because the bunny video has 20% more frames than the bikes video.

SPDER would perform even better than what we present in these results. Training for more steps or using a larger network would also likely lessen the impact of these outliers.

We acknowledge that, ideally, SPDER should not have problems with any samples (however anomalous). Studying these challenging samples may be the basis for future work.

## A.7 VIDEO REPRESENTATION AND FRAME INTERPOLATION

First, we trained a SPDER on a portion of Big Buck Bunny, a standard video used in computer vision research. SPDER achieved a peak PSNR of 29.74 within 450 steps, indicating good quality reconstruction. Note how SPDER's representation also serves a lossy compression scheme because the network has ∼12 million parameters, but the original video has ∼20 million. Second, we trained a 256x256 resolution clip from `skimage.data.bikes()` and it achieved a PSNR of 30.9 within 450 steps. The performances are plotted in Fig. 16.

For the frame interpolation experiments, we trained on the bikes video at 30 FPS, and then asked the model to predict what it would look it if it had twice as many frames (i.e. 60 FPS).

### A.7.1 VIDEO REPRESENTATION EXAMPLES

The examples, from `skvideo.datasets.bikes()`, are in Fig. 17.

### A.7.2 FRAME INTERPOLATION EXAMPLES

Examples are in Fig. 18 and Fig. 19.

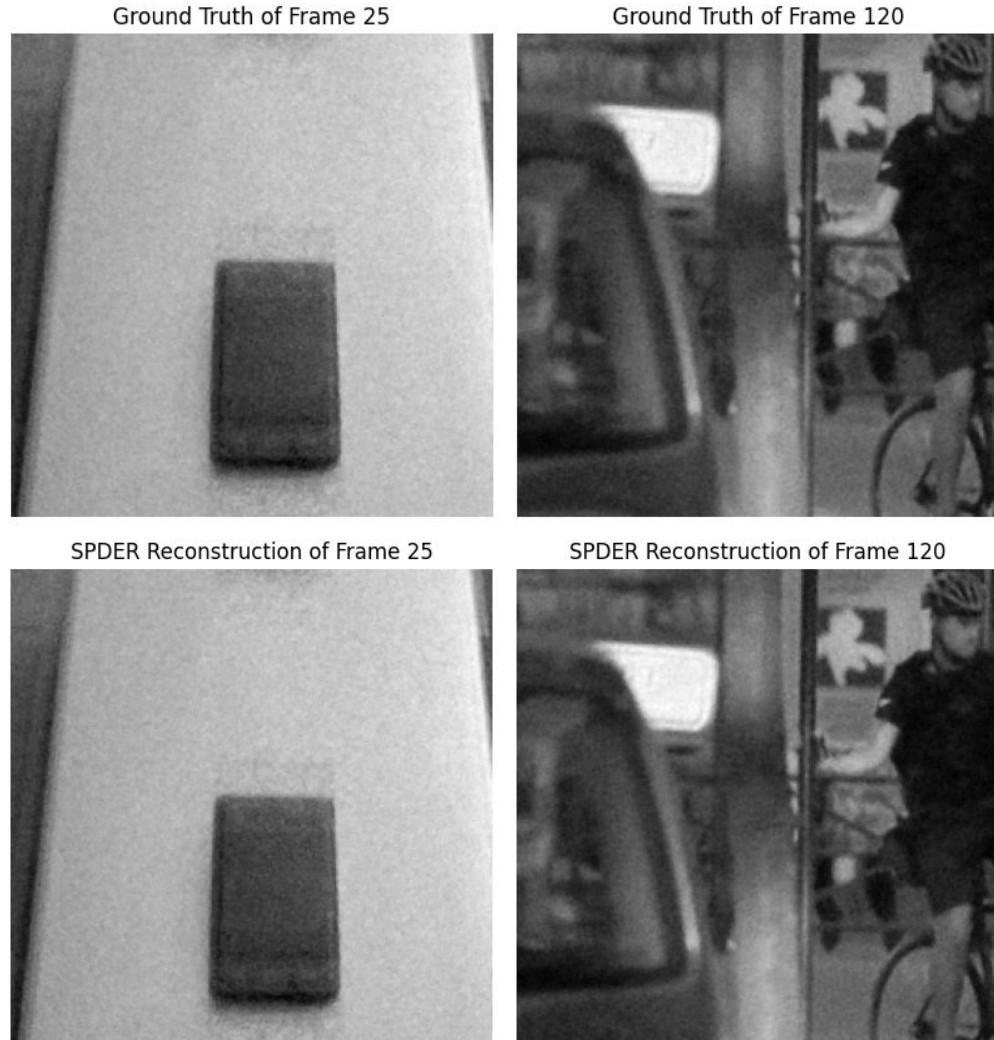

Figure 17: This 250-frame video from `skvideo.dataset.bikes()` is fairly chaotic with many moving parts, yet SPDER is still able to capture it without any artifacts. Here, we chose two frames with overall low and high frequencies (left and right, respectively) to contrast. SPDER evidently has no bias and can encode both accurately.

Ground Truth of Frame 199      Ground Truth of Frame 200

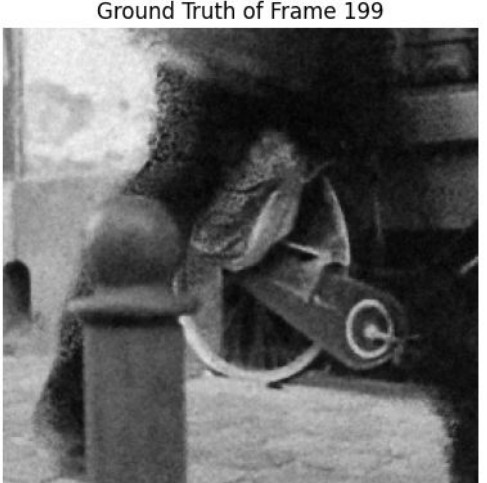 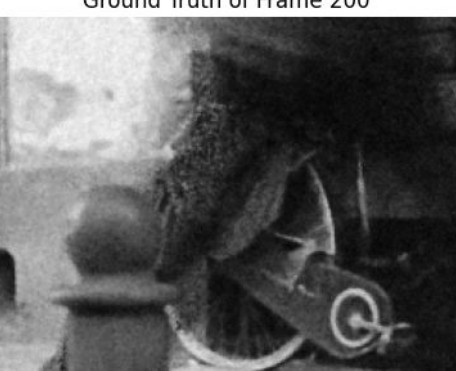

SPDER Reconstruction of Frame 199.5

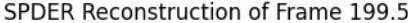

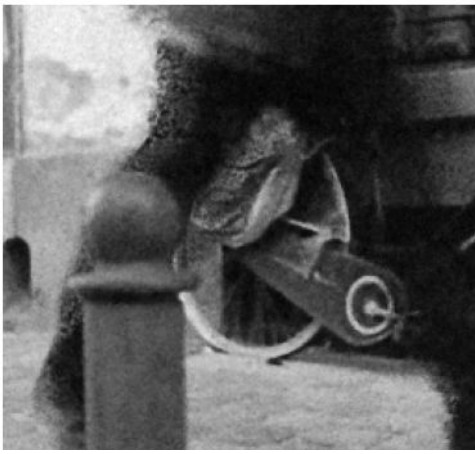

Figure 18: We selected two consecutive frames where movement would be as perceptible as possible. We can see a person's leg move partially behind a pole in the ground truth frames (top left and top right). SPDER's interpolation for the intermediate frame is indeed plausible and doesn't include any visual distortions not already present in the ground truth.

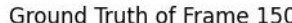
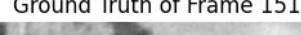
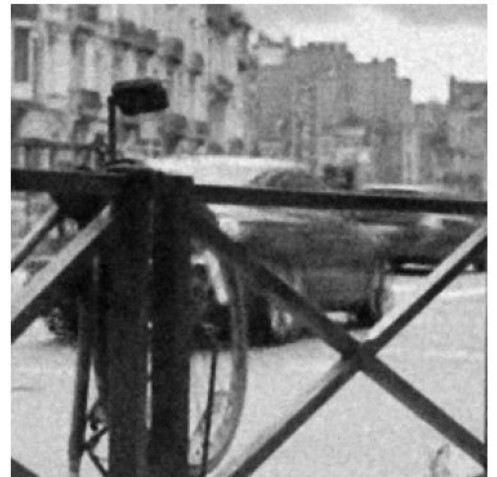
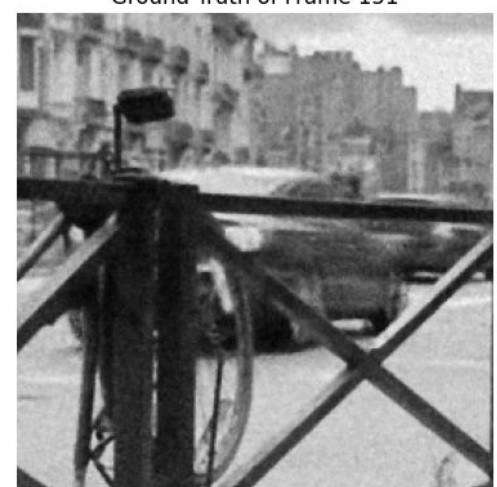
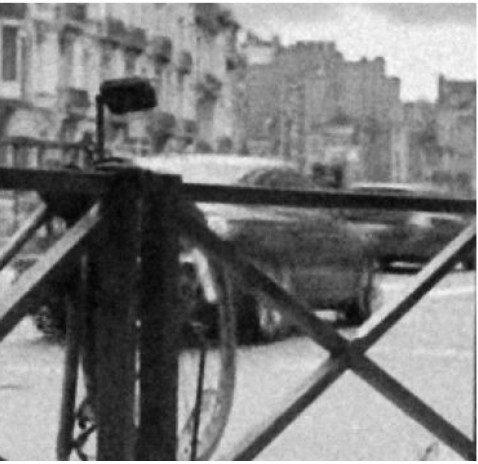

Figure 19: We selected two relatively still consecutive frames and demonstrate how SPDER's interpolation does not seem visually grainier or blurrier (i.e., no higher or lower frequencies introduced) than in frames from the ground truth distribution.

## A.8 COMMENTS ON INSTANT-NGP

Given the hierarchical setup and CUDA C++-tailored implementation, Instant-NGP has more complexity and a significantly large overhead compared to SPDER. We tried running Instant-NGP on several machines, but due to insufficient system requirements, all but a machine with 8 Quadro RTX 6000's (costing over $20,000) failed to run it. Although it offers an impressive speedup compared to other architectures, we found the system requirements to be quite prohibitive compared to using a simple model like SPDER.

That being said, it is also much more efficient and extremely useful for most high-information media. For example, it was much faster than SPDER during training, taking about $\sim$5 seconds for 500 steps on a single 256x256 image, whereas SPDER took $\sim$15 seconds on the same machine. Likewise, it allows for training on objects such as gigapixel images within minutes, which would otherwise take a few hours with the current iteration of SPDER.

To the best of our understanding, Instant-NGP learns a piecewise-linear approximation of the signal and is prone to pseudo-random hash collisions (Müller et al., 2022), so although it achieves very low loss, it technically cannot model the high-frequency detail as well as SPDER. We were unable to retrieve its reconstructed image and therefore could not calculate $\rho_{AG}$ or use it for super-resolution, so we emphasize that any claims we make regarding Instant-NGP's frequency alignment are purely based on our theoretical understanding of it.

We believe that Instant-NGP attempts to overcome spectral bias by having many learned parametric encodings which are responsible for a very small receptive field in the image. Each field has relatively low complexity, which means they can get away with using augmentations like ReLU and bilinear interpolation to approximate the signal piecewise-linearly. However, in SPDER, each neuron has the receptive field of the *entire image*, which means in order to represent it well, it is forced to learn the true frequency structure. To the best of our knowledge, our architecture is the only one that naturally learns the true frequency structure of an image and overcomes spectral bias—notably without hardcoded inductive biases like hierarchies. That being said, we believe combining a hierarchical approach with SPDER is the next reasonable step.

## A.9 REPRODUCIBILITY

By default, we use a 5-layer network with 256 neurons in each layer and a learning rate of $1 \times 10^{-4}$. Input and output values are scaled to be in $[-1, 1]$. As we wish to completely "overfit" to the training data, we use full batch gradient descent (i.e., we do not want the noise from SGD) on each object. We discovered that clamping the input of $\sin(x) \cdot \sqrt{|x|}$ to a tiny value (less than $1 \times 10^{-30}$) avoided division by zero errors on the GPU while maintaining overall functionality. For any resizing operations, we use PyTorch's built-in `torchvision.transforms.Resize` module which uses bilinear interpolation under the hood.

We used two GeForce GTX 1080 Ti's with 12 GB of memory each. We note that all SPDER experiments can be run on personal laptops and require no tailored hardware.

Please refer to the full source code for reproducing the experiments in the supplementary material.

### A.9.1 WEIGHT INITIALIZATION

We followed the weight initialization scheme from Sitzmann et al. (2020). Weights are initialized from $\mathcal{U}\left(-\sqrt{\frac{6}{\text{fan\_in}}}, \sqrt{\frac{6}{\text{fan\_in}}}\right)$, where fan_in is the number of inputs to a layer. They claimed this leads to the input of each activation being normally distributed and keeps gradients stable, which we indeed observed in SPDER (Fig. 8).

Also consistent with their implementation, we multiply the input into each activation by a predetermined $\omega_0$, which is always 30.

### A.9.2 IMAGE REPRESENTATION

Recall that for INRs, the training data are the pixels from a *single* image; the model's goal is to overfit it. The training process can be viewed as an "encoding" step, where the data from an image is loaded

onto the weights of the model. The "decoding" step is simply a single forward pass of the model, where the information from the weights is mapped onto a prediction at each pixel coordinate.

For these experiments, we sampled images from DIV2K, a high-quality super-resolution dataset. Each image was resized to a resolution of 256x256 and fed into an untrained network, which would then train on it for a fixed number of steps. For simplicity, only one channel of each image was included, but note that representing all three is trivial. The loss, peak signal-to-noise ratio (PSNR), and $\rho_{AG}$ between the overfit model's representation after training and ground truth were recorded for training steps 25, 100, and 500. Loss is the mean-squared error between the model's prediction compared to the ground truth over each pixel. Recall that the values come from a scale of $[-1, 1]$ (A.13). PSNR is calculated through Eq. (23). $\rho_{AG}$ comes from Eq. (5) and is simply the cosine similarity between the amplitude spectrum of the model's reconstruction and the ground truth image.

For Instant-NGP, we had to use a much more high-end machine (A.8) to collect data. Note that for ReLU with FFN, the method requires manual tuning of a scale hyperparameter, so perhaps a configuration better than a standard Gaussian exists, but it was not apparent to us even after several attempts. SPDER, on the other hand, requires no such manual tuning for different datasets.

### A.9.3 GRADIENT REPRESENTATION

For gradient representation, we overfit a network to an image using the same hyperparameters as before. The gradient of the reconstructed image is then computed. Specifically, we use the PyTorch function `torch.autograd.grad` to calculate the gradient of the output of the model (i.e., the reconstructed pixels) with respect to the coordinate inputs.

### A.9.4 SUPER-RESOLUTION

For these experiments, we utilized three datasets – DIV2K, Flickr2K, and Flickr-Faces-HQ. DIV2K, a standard benchmark for image super-resolution, comprises 2K resolution RGB images. Flickr2K is a diverse dataset of high-resolution images sourced from Flickr, widely used for training super-resolution models. Lastly, Flickr-Faces-HQ (FFHQ) is a dataset of high-quality PNG images at 1024x1024 resolution, employed to test model performance on detailed, structured imagery like human faces. We test images from each dataset on three super-resolution factors (SRFs) – 2, 4, and 8. For each combination of SRF, dataset, and architecture, we use $N = 12$ images from each dataset.

For training on each dataset, we first sample an image, select a single channel, and then resize it down to 512x512 (the base resolution). We overfit the model (either SIREN or SPDER) to the resized image by training it for 100 steps (by when both models' reconstructions reasonably converge). Using the trained model overfit to the base resolution version of the image, we then ask the model to predict what the image would look like at a resolution that is $2\times$, $4\times$, or $8\times$ that of the base resolution. Recall this is possible because INRs learn a *continuous* representation for an object, and can therefore be queried at any input coordinate value within a reasonable domain. To compute scores on our metrics, we take the original image along the single channel and (if needed) resize it at the given super-resolution to get the ground truth. We calculate the metric between the model's prediction and the ground truth at the given super-resolution, using the same procedure as for that of image representation.

### A.9.5 AUDIO REPRESENTATION

Consistent with the implementation by Sitzmann et al. (2020), we 1) scale the input grid to have bounds $[-100, 100]$ instead of $[-1, 1]$, which is tantamount to scaling the weights of the input layer by 100, and 2) use a learning rate of $5 \times 10^{-5}$.

Each training sample was cropped to the first 7 seconds of clips from ESC-50, a labeled collection of 2000 environmental audio recordings, and then trained on for 1000 steps.

### A.9.6 AUDIO INTERPOLATION

For each ground truth, we sample a clip from ESC-50 and crop it to the first 4 seconds. We train the network (either SIREN or SPDER) on an $8\times$ downsampled version of this ground truth clip for 250 steps. For inference, we then take the original ground truth clip and downsample it by $4\times$, $2\times$, and

$1\times$. Note the overfit network will have seen only half of the first set of values, a quarter of the second, and an eighth of the third during training. The model then outputs a prediction for the ground truth audio clips across these resolutions, and the MSE is recorded. For each combination of architecture and resolution, we use $N = 30$.

Unless stated otherwise, the rest of the implementation is the same as that of for audio representation.

### A.9.7 VIDEO REPRESENTATION

We used a 12-layer 1024-neuron wide network with a learning rate of $5 \times 10^{-6}$ for our video experiments. Each architecture (SIREN and SPDER) was trained for at least 350 steps, and the loss was recorded at each step. To speed up training, we selected a single channel from each video and center-cropped then resized to 256×256. Training on a single video for SPDER took ∼6 hours on our setup with 24 GB of compute.

The first video we trained on was from `skvideo.datasets.bigbuckbunny()`, a standard video used in computer vision research. The first 300 frames were kept. The second video was from `skvideo.datasets.bikes()`. The first 250 out of 300 total frames were kept.

### A.9.8 VIDEO FRAME INTERPOLATION

We used the SPDER trained on the 250-frame video from `skvideo.datasets.bikes()` from the representation experiments. While the original video was shot in 30 FPS, we asked the SPDER to predict what it would look like at 60 FPS by inputting the same coordinate grid but $2\times$ denser along the frame dimension. We then visualized the results.

Unless stated otherwise, the rest of the implementation is the same as that of for video representation.

### A.10 LIMITATIONS

- SPDER shows the most success on tasks where the input comes from a coordinate grid. For instance, it converges slower than NeRFs for novel view synthesis, presumably due to the non-affine nature of the view direction input. Future work can explore preprocessing approaches to aptly project such inputs onto a coordinate space to potentially see the same success as for images.

- When compared to SIREN, SPDER's training is ∼1.3× slower since it must calculate the damping factor for each input. While storing the gradients does consume ∼1.5× more memory, this issue can be circumvented through batch processing.

- Determining the optimal $\delta$ for media types other than image, audio, and video may require some experimentation.

### A.11 IMAGE FFT FORMULA

In the main paper, we included the general formula for the Fourier amplitude spectrum of a discrete-time signal for a vector input. However, the Fourier transform can also apply to tensors, which is relevant in the case for images. Here, we include the formula for the Fourier transform of a 2D tensor:

$$\mathcal{F}(u, v) = \frac{1}{MN} \sum_{x=0}^{M-1} \sum_{y=0}^{N-1} f(x, y) e^{-i2\pi \left( \frac{ux}{M} + \frac{vy}{N} \right)} \tag{18}$$

which we used when evaluating the frequency similarity of images. In this formula, $\mathcal{F}(u, v)$ represents the Fourier transform of the 2D array $f(x, y)$ with respect to the spatial frequencies $u$ and $v$. The size of the array is $M$ by $N$. The summation is taken over all possible values of $x$ and $y$ in the array. The exponential term represents the phase shift at each point in the array. Note that the method `numpy.fft.fft2` can be used to calculate this efficiently, so we included this just for reference.

### A.12 ABLATION ON VARIOUS $\delta$'S

We tested various forms of $\delta$ on the image from `skimage.data.camera()` over 250 steps in Fig. 20. The log-loss results are shown. Using a $\delta(x)$ of ReLU$(x)$ or just $x$ performs extremely poorly,

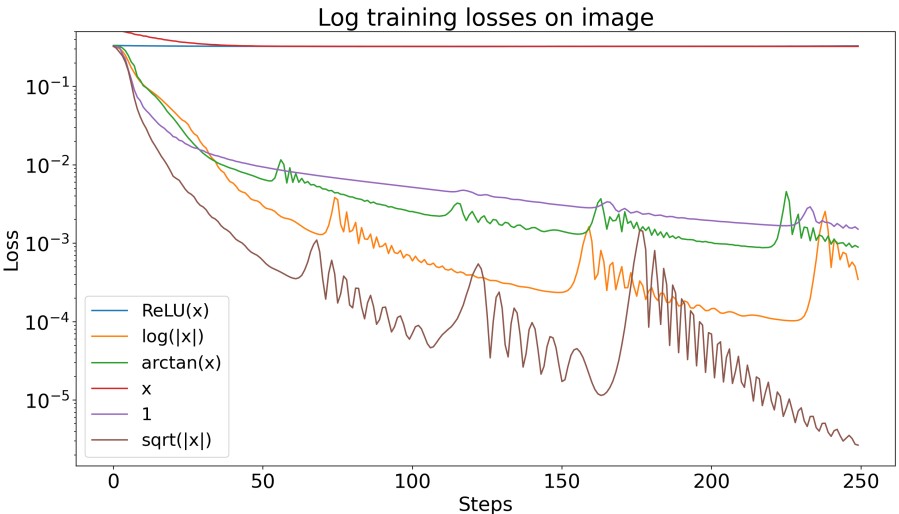

Figure 20: Log training losses on the cameraman image for various $\delta$'s

and the loss curves are cropped off from the top because the losses are so high. The performance of the other $\delta$'s in order from best to worst is $\sqrt{|x|}$, $\log(|x|)$, $\arctan(x)$, and then 1 (which is simply a sinusoidal, used in SIREN).

The loss curves in Fig. 20 for SPDER and SIREN look different from Fig. 1 because we trained on a CPU with no clamping here, but the overall trend remains.

### A.13 CORRESPONDENCE TO 8-BIT SCALE

In our training setup, inputs and outputs are scaled to be in $[-1, 1]$. The mean-squared error (MSE) loss is calculated on this scale as well. To interpret this on an 8-bit scale (where pixel values are integers, $p$, with $p \in \{0, 1, \ldots, 255\}$), we need to apply the following transformation on an output value $y \in [-1, 1]$:

$$f : y \to p; \quad f(y) = \frac{(y + 1) \cdot 255}{2} \tag{19}$$

In practice, however, the bounds of the prediction values slightly diverge from $[-1, 1]$, so the most reliable way to reconstruct images is through normalization:

$$f : y \to p'; \quad f(y) = \frac{(y - y_{min}) \cdot 255}{(y_{max} - y_{min})} \tag{20}$$

If we want to calculate the MSE on the 8-bit pixel scale, we apply this transformation:

$$f : MSE_y \to MSE_p; \quad f(MSE_y) = MSE_y \cdot \left(\frac{255}{2}\right)^2 \tag{21}$$

where we assume $y_{max} - y_{min} \approx 2$. In Table 1, we demonstrate how SPDER has an average training loss of $6.7 \times 10^{-8}$, which corresponds to an MSE of 0.001 pixels$^2$ on an 8-bit pixel scale, so incorrectly predicted bits are extremely rare. No reconstruction at this resolution had any artifacts, according to our observations.

### A.14 PSNR CALCULATION

The formula for peak signal-to-noise ratio (dB) is given by:

$$PSNR = 10 \cdot \log_{10}\left(\frac{MAX_I^2}{MSE}\right) \tag{22}$$

Note that outputs are scaled from $[-1, 1]$ and have a range of 2. Using the same MSE in training, we can calculate PSNR as follows:

$$10 \cdot \log_{10} \left( \frac{(1 - (-1))^2}{\text{MSE}} \right) = 10 \cdot \log_{10} \left( \frac{4}{\text{MSE}} \right) \tag{23}$$

Therefore, PSNR can be directly determined from the training loss (MSE) on each sample. We only include it because of its interpretability.

