# OpenReview forum: "SPDER: Semiperiodic Damping-Enabled Object Representation"
_ICLR.cc/2024/Conference — ICLR 2024 poster_

### Official Review · Reviewer_TKxh · 2023-10-29

**Soundness:** 2 fair
**Presentation:** 4 excellent
**Contribution:** 3 good
**Rating:** 5
**Confidence:** 4

**Summary:**

The paper proposes a novel architecture for coordinate-based networks used for signal representation. The key contribution of this architecture is that the activation function is changed to a sinusoidal function multiplied by a damping function. This result leads to improvement over existing activations which either use ReLU, and have a spectral bias towards low-frequency functions, or use sinusoidal activations, which do not have a notion of locality due to the lack of damping factor and thus lead to poorer reconstructions. This contribution is justified empirically by showing that it leads to significantly better signal fits on image signals.

**Strengths:**

In my opinion, the strengths of the paper are:
1. The paper is presented exceptionally clearly. The activation function and possible choices are explained very well, and the reasoning for the chosen function is justified intuitively, and there are theoretical guarantees on the Lipschitz constant of the function represented by the network. I view this as very valuable as the paper can have more impact if researchers understand the reason for the increase in performance, as this is under-explored area of activation functions in coordinate-based networks.
2. The comparisons show that the proposed SPDER activation function leads to significant improvement over positional encoding, SIREN, and ReLU networks in signal memorization tasks. This shows that the relatively simple contribution does lead to an improvement in what is claimed, and has the potential to be impactful.

**Weaknesses:**

In my opinion, the weaknesses of the paper are:
1. I think that additional experiments could be used as there are other coordinate-based network architectures which claim improvements over positional encoding and SIREN architectures. I'm not sure why Instant-NGP is chosen here, as it is a hybrid implicit-explicit representation and is very different from the other coordinate-based networks. I believe a better fit would be [1] or [2], which show that simple networks with different activations, structures, or positional embeddings can improve on the FF/SIREN architectures.
2. Following the additional experiments thread, I'm not sure why comparisons on radiance field fitting are not included. While interesting as a toy problem, image signal representation is not very useful in many computer vision tasks, besides maybe super-resolution. However, coordinate-based networks have been very useful in computer graphics, where they can be used in solving inverse problems and reconstructing 3D from 2D. Not including an experiment on radiance field, which is the most impactful field where an improvement in coordinate-based network fitting would be utilized, leads me to believe that the impact of the method is severely limited due to some unknown reason, perhaps speed or memory constraints in the architecture?
3. Even simpler than radiance field fitting, I don't understand why there is no experiment on overfitting 3D shapes, such as SDFs? This is where coordinate-based networks originated, and could be used as a compressive representation. The paper mentions compressive signal representations in the conclusion, but there is no study on how effective SPDER is for this - no study on memory consumption, or size of networks needed to reach an acceptable level of quality. This is usually done when trying to represent 3D objects with SDFs, so this leads me to believe the method may have some limitations here.

[1] https://openreview.net/forum?id=OmtmcPkkhT

[2] https://arxiv.org/abs/2106.01553

**Questions:**

I do not have any additional questions on the paper, as it is described very clearly. Overall, I think that the contribution of the paper is strong due to its simplicity and demonstrated improvement on image-signal overfitting tasks. However, I think that the evaluations are limited - there are other coordinate-based network architectures which lead to improvement over the baselines which are compared to here, and most importantly, the method is not compared on radiance fields, which is the most impactful application of coordinate-based networks. I think that addressing these two weaknesses would make the paper significantly stronger and in that case I would be happy to raise my score.

**Update after author response**

After reading the author response, I am not inclined to change my score. While I appreciate the additional comparisons, and I believe the paper does a convincing job showing that overfitting image signals can be done extremely well with SPDER, I think the paper weakness is justifying *why* this is a relevant problem. The method seems to not work on radiance fields, has not been shown to work on SDFs, nor am I sure it offers compression benefits in overfitting image signals. Thus, I'm not sure what application it would be used in. On the other hand, there is no theoretical proof that it is able to encode a wider frequency band of signals, only empirical results on a relatively narrow range of experiments. With further work, I believe the paper could make a meaningful contribution, but in this state I remain borderline.

---

> ### Author Response · Authors · 2023-11-21
>
> We thank the reviewer for their thoughtful feedback on our submission. Please find our answers and responses below.
>
> **> I think that additional experiments could be used as there are other coordinate-based network architectures which claim improvements over positional encoding and SIREN architectures. […] I believe a better fit would be [1] or [2], which show that simple networks with different activations, structures, or positional embeddings can improve on the FF/SIREN architectures.**
>
> Following the reviewer’s suggestion, we ran additional experiments applying SPDER activations to the models from both of the suggested baselines: GaborNet from MFN [1] and Spline Positional Encoding [2]. Then, we evaluate these new models on the same image dataset as the original MFN paper [1].
>
> **PSNRS**
> | Training Steps | 5-layer SPDER | 5-layer GaborNet | Spline | 3-layer SPDER | 3-layer GaborNet |
> |----------------|---------------|------------------|--------|---------------|------------------|
> | 500            | 75.32         | 49.14            | 29.73  | 33.97         | 46.59            |
> | 1000           | 78.01         | 56.07            | 38.87  | 38.62         | 52.78            |
> | 5000           | 78.01         | 69.71            | 44.40  | 41.44         | 67.08            |
> | 10000          | **108.40**        | 77.65            | 65.86  | 41.44         | 71.98            |
>
> To highlight the difference, SPDER beats the best GaborNet loss in 800 steps and the best Spline loss in 300 steps. SPDER’s representation is lossless because the loss is so small it doesn’t show up on an 8-bit scale (not true for the others, see Appendix A.13). The 5-layer SPDER (which we tested in our paper) has significantly higher PSNRs than the others in very few training steps, meaning it is also very efficient.
>
> We will be sure to include these results in the main paper.
>
> **> I'm not sure why Instant-NGP is chosen here, as it is a hybrid implicit-explicit representation and is very different from the other coordinate-based networks.**
>
> Its paper says it is “state-of-the-art quality on a variety of tasks” related to image representation. We wanted to show our simple MLP can beat this complicated setup which requires a heavy GPU setup.
>
> Also stated in Appendix A.8, our method could also be combined with a hierarchical one like Instant-NGP, as it is more efficient.
>
> **Comparisons on radiance fields and SDFs**
>
> We observed that SPDER works for signals that can be well approximated in a low dimensional representation over a frequency domain. For example, an image can be well approximated by using the discrete cosine transform (JPEG). Similar methods exist for audio and video. To the extent of our knowledge, domains like SDFs or radiance fields do not exhibit a similar property. We hypothesize that this fact renders SPDER ineffective in these domains (mentioned at the bottom of page 4). This is likely an instance of the “no free lunch” theorem; no single architecture can perform the best on all modalities.
>
> We anticipate there can be future work on projecting these modalities into a vector space where SPDER’s activation functions can perform effectively. We believe this is reasonable as current methods for SDFs and radiance fields are generally divided into two parts: 1) preprocessing/conditioning 2) MLP. Our current implementation only uses the latter.
>
> The main claim of our paper is that existing models are prone to the inductive bias of not being able to encode all frequencies, which we show our simple MLP overcomes. This is due to the “damping factor” in the nonlinearity (i.e. the square root in $\sin(x) * \sqrt{|x|})$ which allows the network to remember where inputs came from. This is most evident in the image case.

---

### Official Review · Reviewer_9hcy · 2023-10-31

**Soundness:** 3 good
**Presentation:** 3 good
**Contribution:** 2 fair
**Rating:** 8
**Confidence:** 3

**Summary:**

This paper proposes a novel family of activation functions to enable an MLP to represent an input signal (image, audio) to very high fidelity. Both the loss and the architecture are modified to enable this. A number of experiments are provided to show the benefits of the resulting model.

**Strengths:**

The paper is simply written and easy to follow (although slightly prone to hyperbolic language; pls see suggestion below). The provided experiments, while not comprehensive, paint a solid picture of the abilities of the model to represent signals in near-lossless manner. The method itself seems very simple to code (although no code was provided, and no training details given). Overall the paper has the strengths of simplicity and should be very easy to use in practice.

**Weaknesses:**

1. I am concerned that this method tends to overfit heavily and easily to a given particular input image/signal, and evidence for generalization was very limited to (as far as I can tell) a single video example. How sensitive is the model to noise in the input, for example? This can be important for real-world applications.

2. I am not an expert in this space, but one question I have is whether the baselines are strong. SIREN is the most commonly used baseline. Maybe the authors can provide some perspective in this regard.

Lastly, I would recommend the authors, in the spirit of academic inquiry, to tone down the language a little bit. An example is a statement of the form: "We demonstrate how SPDER not only has incredibly low loss", can be restated as "We demonstrate how SPDER not only has significantly lower loss" etc. This is a minor point, admittedly my personal taste.

**Questions:**

There is a statement buried in page 4 that says: "For example, they are less successful at novel view synthesis, where the input direction is polar.". However, no examples of performance on novel view synthesis was provided. Can the authors comment on this?

---

> ### Author Response · Authors · 2023-11-21
>
> We thank the reviewer for their thoughtful feedback on our submission. Please find our answers and responses below.
>
> **> The method itself seems very simple to code (although no code was provided, and no training details given). Overall the paper has the strengths of simplicity and should be very easy to use in practice.**
>
> The code was provided in the supplementary material, along with the appendix. Training details are included in Appendix A.9 (Reproducibility).
>
> **> There is a statement buried in page 4 that says: "For example, they are less successful at novel view synthesis, where the input direction is polar.". However, no examples of performance on novel view synthesis was provided. Can the authors comment on this?**
>
> We observed that SPDER works for signals that can be well approximated in a low dimensional representation over a frequency domain. For example, an image can be well approximated by using the discrete cosine transform (JPEG). Similar methods exist for audio and video. To the extent of our knowledge, domains like SDFs or radiance fields do not exhibit a similar property. We hypothesize that this fact renders SPDER ineffective in these domains. This is likely an instance of the “no free lunch” theorem; no single architecture can perform the best on all modalities.
>
> The main claim of our paper is that existing models are prone to the inductive bias of not being able to encode all frequencies, which we show our simple MLP overcomes. This is due to the “damping factor” in the nonlinearity (i.e. the square root in $\sin(x) * \sqrt{|x|})$ which allows the network to remember where inputs came from. This is most evident in the image case.
>
> **> I am concerned that this method tends to overfit heavily and easily to a given particular input image/signal, and evidence for generalization was very limited to (as far as I can tell) a single video example. How sensitive is the model to noise in the input, for example? This can be important for real-world applications.**
>
> The implicit neural representations should “overfit” to the signal or the object it is being trained on, as the goal is essentially to learn a mapping from pixel position to pixel value for a single object. The desired generalization is *between* the pixels in an image or between frames in a video. Our experiments demonstrate this generalization effect in the image superresolution task (Section 4.3, Appendix A.5), audio interpolation (Section 4.4, Appendix A.6), and video interpolation (Section 4.5, Appendix A.7).
>
> Regarding video examples, we report on two videos (not one). They are “Big Buck Bunny” and “Bikes”, which are from skvideo and are standard in video computer vision research (Appendix A.7).
>
> **> I am not an expert in this space, but one question I have is whether the baselines are strong. SIREN is the most commonly used baseline. Maybe the authors can provide some perspective in this regard.**
>
> We compared SPDER to the most popular coordinate-based MLP models we found in the existing literature. Most of these models are simple neural networks. We also compared our work to Instant-NGP (Section 4.2 Image Representation), which uses a much more complicated setup and is state-of-the-art for many INR tasks.
>
> **> Lastly, I would recommend the authors, in the spirit of academic inquiry, to tone down the language a little bit.**
>
> We will change the wording accordingly.

---

> > ### Comment · Reviewer_9hcy · 2023-11-21
> > **Thank you for addressing my concerns**
> >
> > I have upgraded my score accordingly.

---

### Official Review · Reviewer_V9Ce · 2023-11-01

**Soundness:** 2 fair
**Presentation:** 3 good
**Contribution:** 3 good
**Rating:** 6
**Confidence:** 4

**Summary:**

This paper proposed a simple architecture to overcome the spectral bias towards lower frequencies in traditional neural network.
It formulated a so called damping function implemented by MLP. It contains an activation function composed of a sinusoidal multiplied by a sublinear function.
The sinusoidal enables the network to automatically learn the positional encoding of an input coordinate while the damping passes on the actual coordinate value by preventing it from being projected down to within a finite range of values.
Further experiments demonstrated good performance on multiple downstream tasks such as image super-resolution and video frame interpolation.

**Strengths:**

+ The proposed method is novel.
+ The introduced damping function is interesting and theoretical sound.
+ The paper shows cases of multiple potential applications using proposed method.
+ Experiments demonstrated that it can largely improve the performance and training efficiency in dummy setups.

**Weaknesses:**

- Do not prove the derivatives of different variants of proposed method.
- It lacks experiments on more realistic datasets.

**Questions:**

Can you evaluate the proposed method on more complex images? Consider to perform formal evaluations on popular benchmarks, such as CelebA, UHDSR4K and Vimeo-90k to demonstrate its superiority.

---

> ### Author Response · Authors · 2023-11-21
>
> We thank the reviewer for their thoughtful feedback on our submission. Please find our answers and responses below.
>
> **> Do not prove the derivatives of different variants of proposed method.**
>
> We prove how activation functions of the form $\sin(x) * \delta(x)$ for any sublinear $\delta(x)$ suffice for our task in Appendix A.1 (Proof of Lipschitz Bound).
>
> **> Can you evaluate the proposed method on more complex images? Consider to perform formal evaluations on popular benchmarks, such as CelebA, UHDSR4K and Vimeo-90k to demonstrate its superiority.**
>
> We report our results on the DIV2K dataset (“DIVerse 2K resolution high-quality images”—commonly used for super-resolution and representing complex images) in Table 1. In fact, it is typical for the signal representation literature to evaluate only a few samples [1, 2, 3]. Our sample size is 800, which is far more than the largest one [3] with N=32.
>
> [1] Sitzmann et al., Implicit Neural Representations with Periodic Activation Functions, NeurIPS 2020.
> [2] Fathony et al., Multiplicative Filter Networks, ICLR 2021.
> [3] Tancik et al., Fourier Features Let Networks Learn High Frequency Functions in Low Dimensional Domains, NeurIPS 2020

---

### Meta-Review · Area_Chair_RwT6 · 2023-12-12

**Metareview:**

Prior work on implicit neural representations is biased towards fitting lower frequencies. In order to mitigate that issue, this submission proposes an activation function composed of a sinusoidal multiplied by a sublinear damping function. The authors have provided theoretical and empirical evidence suggesting that their proposed method outperforms prior work in fitting to functions, especially those in the image domain.
The reviewers appreciate the simplicity of the method and observations, and the quality of the results.
However, one reviewer raised concerns that the applicability of the propose method might be very narrow, and only suited for the image/audio domain, without good results on fitting to the 3D domain (SDFs/radiance fields). However, all 3 reviewers agree that the submission is well-written, and that the method is simple and effective, even though probably narrow in applicability. Given this background, this submission does provide useful information to its readers and possible motivation towards more general future work. The authors are encouraged to improve their submission using feedback from the reviewers; perhaps even add a comparison on fitting to 3D .

**Justification For Why Not Higher Score:**

The applicability of the proposed method might be limited.

**Justification For Why Not Lower Score:**

All reviewers agree that this submission has content of value to the audience, despite its simplicity.

---

### Decision · Program_Chairs · 2024-01-16

Accept (poster)